# Characterizing permafrost active layer dynamics and sensitivity to landscape spatial heterogeneity in Alaska

Yonghong Yi[1]*, John S. Kimball[1], Richard H. Chen[2], Mahta Moghaddam[2], Rolf H. Reichle[3], Umakant Mishra[4], Donatella Zona[5], Walter C. Oechel[5]

[1]Numerical Terradynamic Simulation Group, The University of Montana, Missoula MT, USA
[2]Department of Electrical Engineering, University of Southern California, CA, USA
[3]Global Modeling and Assimilation Office, NASA Goddard Space Flight Center, Greenbelt, MD, USA
[4]Environmental Science Division, Argonne National Laboratory, Argonne, IL, USA
[5]Department of Biology, San Diego State University, San Diego, CA, USA

*Correspondence to*: Yonghong Yi (yonghong.yi@ntsg.umt.edu)

**Abstract.** An important feature of the Arctic is large spatial heterogeneity in active layer conditions, which is generally poorly represented by global models and can lead to large uncertainties in predicting regional ecosystem responses and climate feedbacks. In this study, we developed a spatially integrated modelling and analysis framework combining field observations, local scale (~ 50 m resolution) active layer thickness (ALT) and soil moisture maps derived from airborne low frequency (L+P-band) radar measurements, and global satellite environmental observations to investigate the ALT sensitivity to recent climate trends and landscape heterogeneity in Alaska. Modelled ALT results show good correspondence with in situ measurements in higher permafrost probability (PP $\geq$ 70%) areas (n = 33, R = 0.60, mean bias = 1.58 cm, RMSE = 20.32 cm), but with larger uncertainty in sporadic and discontinuous permafrost areas. The model results also reveal widespread ALT deepening since 2001, with smaller ALT increases in northern Alaska (mean trend = 0.32 $\pm$ 1.18 cm yr$^{-1}$) and much larger increases (> 3 cm yr$^{-1}$) across interior and southern Alaska. The positive ALT trend coincides with regional warming and a longer snow-free season (R = 0.60 $\pm$ 0.32). A spatially integrated analysis of the radar retrievals and model sensitivity simulations demonstrated that uncertainty in the spatial and vertical distribution of soil organic carbon (SOC) was the largest factor affecting modeled ALT accuracy, while soil moisture played a secondary role. Potential improvements in characterizing SOC heterogeneity, including better spatial sampling of soil conditions and advances in remote sensing of SOC and soil moisture, will enable more accurate predictions of active layer conditions and refinement of the modelling framework across a larger domain.

## 1 Introduction

Regional warming in the northern high latitudes is occurring at roughly twice the global rate, leading to widespread permafrost degradation (Jorgenson et al., 2006; Romanovsky et al., 2010) and substantial changes in hydrologic and ecosystem processes, including earlier and potentially longer growing seasons (Kim et al., 2012), expansion of tundra shrub

cover (Tape et al., 2006), changes in lake and wetland areas (Smith et al., 2005), and increasing thermokarst development (Liljedahl et al., 2016) and fire disturbances (Grosse et al., 2011). Thawing of permafrost can lead to widespread changes in the terrestrial water cycle, including alteration of water storage in surface reservoirs (including lakes, wetlands and ponds) and the active layer (Walvoord et al., 2016). These hydrologic shifts will likely trigger profound changes to almost every aspect of the Arctic biophysical system.

Understanding the linkages between changes in the permafrost active layer and hydrologic and ecological processes is hampered by inconsistent information on active layer properties and dynamics over large regional extents. Traditional estimates of permafrost active layer conditions have relied on detailed ground surveys and measurements from sparse

monitoring sites (Romanovsky et al., 2010; Osterkamp, 2007). More recent attempts have also incorporated ground-based remote sensing such as ground penetrating radar (GPR) and electrical resistivity measurements, but only over limited local extents (Sjöberg et al., 2015; Jorgenson and Grosse, 2016). Several studies have used empirical models driven by in situ ground observations and other geospatial datasets to provide fine-scale (< 100 m resolution) estimates of active layer and near-surface permafrost conditions (Mishra and Riley, 2014; Pastick et al., 2015). However, the accuracy of these methods is

limited by the ability of sparse ground measurements representing landscape heterogeneity, and the resulting empirical models provide only limited insight and mechanistic understanding of underlying processes affecting active layer conditions. Detailed process models have been developed to address the above limitations, while the model accuracy is constrained by a lack of information for effective model parameterization, limited process understanding and coarse spatial scales of regional drivers (Yi et al., 2015; Jafarov and Schaefer, 2016). Particularly, large uncertainties remain in characterizing regional

variability of subsurface soil organic carbon (SOC) content due to limited ground observations of this parameter in the Arctic region (Ping et al., 2008; Burnham and Sletten, 2010) and its effect on ground temperature evolution.

An important feature of permafrost affected soils is the large spatial heterogeneity in permafrost and active layer conditions (Zona et al., 2011), which is generally poorly represented in global models. Capabilities for effective assessment and

monitoring of active layer dynamics at the landscape scale (≤ 1 km) are currently lacking, but are needed to understand processes that govern the permafrost distribution in global carbon and climate models (Slater and Lawrence, 2013; Schuur et al., 2015; Jiang et al., 2016). Satellite remote sensing allows for regional detection and monitoring of surface and subsurface conditions related to active layer properties (Jorgenson and Grosse, 2016), and regionally refined satellite data driven models offer a potential means for regional assessment and monitoring of permafrost active layer properties at suitable landscape

scales. The on-going NASA Arctic-Boreal Vulnerability Experiment (ABoVE) field campaign is collecting a wide range of datasets intended to support regional integration and synthesis of geospatial information and associated data products generated from airborne and spaceborne remote sensing, and detailed ground observations. A major goal of the ABoVE is to develop a modelling framework to improve representation of key processes in the Arctic and boreal landscape, and study

potential climate feedbacks via scaling of local processes to broader spatial extents using multi-scale and multi-sensor remote sensing (Goetz et al., 2011).

Consistent with the ABoVE modelling goal, we developed a spatially integrated process modelling and data analysis framework to characterize regional patterns and recent (2001-2015) changes in active layer thickness (ALT) and underlying environmental controls across Alaska. The framework combines field measurements, local scale (~ 50 m resolution) active layer maps derived from airborne low frequency radar remote sensing, and landscape level ($\geq$ 1 km resolution) environmental observations from global satellite microwave and optical-infrared sensors. Satellite sensor records including land surface temperature (LST) and snow cover extent (SCE) from MODIS (MODerate resolution Imaging Spectroradiometer) were used to drive a detailed 1-D soil heat transfer model, with soil thermal conductivity defined using daily surface and root-zone soil moisture observations from the SMAP (Soil Moisture Active and Passive) mission. The model was used to estimate regional patterns and recent changes in permafrost extent and ALT across Alaska at landscape scale (~ 1 km). A detailed model sensitivity assessment was conducted to determine the major sources of uncertainty in model simulated ALT and the primary factors influencing landscape scale ALT heterogeneity. Local scale ALT and soil moisture maps derived from low frequency (L+P-band) airborne radar backscatter measurements from NASA Uninhabited Aerial Vehicle Synthetic Aperture Radar (UAVSAR) and Airborne Microwave Observatory of Subcanopy and Subsurface (AirMOSS) sensors were used to evaluate the model sensitivity to finer-scale patterns in soil moisture and soil organic fraction.

## 2 Methods

### 2.1 The modelling framework

The model simulations were conducted using a detailed soil process model (Rawlins et al., 2013; Yi et al., 2015) primarily driven by global satellite observation records including land surface "skin" temperature (LST), SCE and surface to root zone ($\leq$ 1 m depth) soil moisture (SM). The soil process model defines up to 23 distinct soil layers down to 60 m below surface. The model uses a numerical approach to solve the 1-D heat transfer equation with phase change included to simulate snow/ground and subsurface thermal dynamics and temperature profiles. The model also accounts for the impacts of SOC content on soil thermal properties. The model was successfully applied to the pan-Arctic region for mapping permafrost extent and active layer dynamics, but at a relatively coarse (~ 25 km) spatial resolution (Yi et al., 2015). In the previous study, global coarse-resolution (~ 0.5°) reanalysis data, including surface air temperature and precipitation, were used as primary model inputs; the model soil thermal properties were regulated by soil moisture content simulated by a water balance model coupled with the soil thermal model (Rawlins et al., 2013). In the current study, however, the satellite-based LST time series were used with snow depth and density data from global reanalysis as major model drivers, with soil thermal properties parameterized using soil moisture profiles from a global data assimilation system.

The soil process model was run at 1-km resolution and 8-day time step consistent with the MODIS LST and SCE inputs (2000-2015). ==The MODIS LST and SCE data were largely affected by clouds in the study area; therefore 8-day temporal composite data were used as the model inputs. Our test runs indicated relatively small differences between model simulated== ==soil temperatures at 8-day and daily time steps (Figure S1).== Primary model inputs (Table 1) included MODIS (Collection 5) 8-day composite 1-km LST (MOD11A2; Wan et al., 2015) and 500-m SCE records (MOD10A2; Hall and Riggs, 2016), SMAP 9-km NatureRun (Version 4) and Level 4 daily surface (≤ 5 cm depth) and root zone (0-1 m depth) soil moisture (L4SM, Reichle et al., 2016), and daily snow depth and snow density from MERRA-2 global (~ 0.5° resolution) reanalysis data (Gelaro et al., 2017). Model simulations were conducted over the Alaskan domain, encompassing an extent of approximately 1.21 million km$^2$. Prior to the model simulations, all datasets were re-gridded to a consistent 1-km Albers equal area projection and 8-day time step for the Alaskan domain. The soil freezing/thawing depth for each 8-day time step was determined as the soil depth crossing the 0°C threshold based on the model simulated soil temperature profile. The ALT was defined as the maximum soil thawing depth throughout the year.

## 2.2 Datasets

### 2.2.1 Model inputs

The MODIS LST and SMAP L4SM products were used to define model boundary conditions and soil thermal properties. MODIS LST was limited to clear-sky conditions, and cold biases were generally found with the MODIS LST data in the Arctic region during the winter season (Westermann et al., 2012). We derived an empirical correction scheme to mitigate the cold bias in MODIS LST data using MERRA-2 total cloud fraction and 2-m air temperature (T2M) data. The MODIS LST data were first aggregated to 0.5° resolution; a linear-regression equation was then derived between MERRA-2 cloud fraction and the difference between MODIS LST and MERRA-2 T2M during the sub-zero period for each biome type. The resulting regression equations were then applied to the original MODIS LST records for the sub-zero period. After bias correction, the MODIS mean LST bias during winter was reduced from -3.66 °C to -0.87 °C relative to in situ air temperature observations in Alaska. The SMAP NatureRun and L4SM surface and root zone soil moisture records were combined to define a continuous soil moisture time series for the entire study period. The SMAP NatureRun soil moisture record was used with the operational L4SM record because the SMAP L4SM operational record doesn't begin until March 2015. The SMAP NatureRun is generated using the same GMAO GEOS-5 land surface scheme and surface meteorology as the L4SM product (Reichle et al., 2017) and showed minimal discontinuity with the L4SM product over the Alaska domain. However, the NatureRun soil moisture product is derived without the benefit of model assimilated SMAP brightness temperature observations.

MERRA-2 daily snow depth and density data were used to account for the effects of seasonal snow cover evolution on the ground thermal regime, with changes in seasonal snow thermal properties derived from snow density (Yi et al., 2015). The soil thermal regime is particularly sensitive to changes in snow cover conditions during snow onset and offset periods, and large-scale reanalysis snow datasets generally have difficulty capturing snow cover spatial heterogeneity, especially during seasonal transition periods (Westermann et al, 2017). Therefore, the MODIS 500-m SCE data were used to adjust the 1-km snow depth and density estimates downscaled from the global reanalysis data (i.e. MERRA-2) during the snow onset/offset period. The snow cover status for each 1-km pixel was defined by choosing the observations that occurred most often based on the 500-m MOD10A2 product. There are substantial areas affected by cloud cover in the Arctic region, especially during the snow season; to minimize cloud effects, pixels identified as cloud contaminated were reclassified as either snow or non-snow covered if the two temporally adjacent periods were both identified as cloud free and indicated consistent snow or non-snow covered conditions. During snow melting and accumulation periods, each coarse MERRA-2 grid cell is generally not fully covered by snow and the MODIS SCE product was used to identify snow-free pixels and adjust the downscaled snow depth/density data for each 1-km pixel within the 0.5° MERRA-2 grid. A more sophisticated downscaling scheme should account for the difference between the MERRA-2 snow cover fraction and MODIS SCE. However, the timing of snow offset/onset derived from the downscaled 1-km MERRA-2 snow depth data showed similar spatial and temporal variations as the MODIS data (Figure S2), indicating that the simple downscaling scheme was generally effective. However, relatively large differences were observed in the timing of snow onset between the MODIS and MERRA-2 records, which was attributed to a greater prevalence of shallow and sporadic snow cover during initial snowpack development in the autumn.

Other ancillary model inputs included the 30-m national land cover database 2011 (Jin et al., 2013), 50-m SOC estimates for Alaska (to 1-m depth; Mishra et al., 2016), and the global 9-km mineral soil texture data developed for the SMAP L4SM algorithm (De Lannoy et al., 2014). The dominant land cover type within each 1-km pixel was used to define the modelling domain, with open water and perennial ice/snow areas excluded from the model simulations. The soil texture and SOC data were used to define the soil properties including thermal conductivities and heat capacities. The SMAP soil texture dataset was generated using multiple soil databases, but primarily used information from the State Soil Geographic (STATSGO2) dataset in Alaska (De Lannoy et al., 2014). The sand and clay fraction data layers of this dataset were resampled to 1-km resolution and used to calculate soil thermal and hydraulic properties for the mineral soils (Lawrence and Slater, 2008). The Alaska SOC map was derived from a geospatial model involving more than 500 soil profile observations and spatial environmental variables, which provides comparable estimates of Alaskan SOC stocks as previous studies, but available at a much finer (50 m) resolution (Mishra et al., 2016). The SOC data was distributed through the top 10 model layers ($\leq 1.05$ m depth) following an exponentially decreasing curve (Jobbagy and Jackson, 2000; Hossain et al, 2015) to calculate the soil carbon fraction of each soil layer as described in Section 2.3.1. The soil physical properties for each soil layer were assumed to be a weighted combination of values of mineral soils and pure organic soils based on the estimated soil carbon fraction following Yi et al. (2015).

## 2.2.2 In situ data

The soil thermal model, as a component of a coupled permafrost hydrology model, was previously validated using in situ soil temperature and soil moisture data from more than 20 Eddy Covariance (EC) tower sites across the pan-Arctic region (Yi et al., 2015). In this study, the model was validated using a limited set of in situ soil temperature measurements from three eddy covariance tower tundra monitoring sites in Alaska (Table S1). The modelled ALT estimates were also validated against in situ ALT measurements from the regional CALM (Circumpolar Active Layer Monitoring) network (Brown et al., 2000). The three tower sites were selected mainly for having relatively good quality surface meteorology and temperature measurements, and supporting information on ground surface conditions. All three tower sites are underlain by permafrost, with relatively large soil organic layer thickness (OLT) and shallow seasonal thaw depth (~40 cm) (Euskirchen et al., 2012; Nakai et al., 2013; Oechel et al., 2014). For the tower site comparisons, the soil process model was parameterized and driven by local tower site meteorological and OLT observations when available. The OLT observations were used to define the depth of the model soil layers with 100% SOC fraction; this simplifying assumption was made in the absence of more detailed SOC profile measurements and to facilitate the model parameterization process. There are ~ 60 in situ CALM sites across the Alaska study domain, with 35 sites located in areas with permafrost probability ≥ 70% estimated from an ancillary permafrost map (Pastick et al., 2015).

## 2.2.3 Airborne Radar retrievals

We conducted an integrated analysis of in situ CALM measurements, soil process model simulations and airborne radar retrievals of soil moisture and ALT over a regional flight transect along the Dalton Highway (DH) in northern Alaska (Figure S3; 148.39-149.05°W, 68.78-70.40°N). The airborne radar retrievals were derived from combined (L+P-band) radar backscatter measurements (~50 m resolution) acquired from coordinated UAVSAR and AirMOSS flights acquired in October 2015, in preparation for the NASA ABoVE campaign. The combination of low-frequency vertically and horizontally polarized P-band (430 MHz) and L-band (1.2 GHz) radar backscatter retrievals provides enhanced sensitivity to active layer conditions, with a greater degree of freedom for distinguishing multiple soil parameters (Du et al., 2015; Chen et al., 2016). The airborne ALT retrievals were derived from the radar backscatter observations using a three-layer (frozen-thawed-permafrost) soil dielectric model, which was parameterized to represent a frozen surface layer overlying a deeper thawed layer for partially frozen conditions in October. The thawed portion of the active layer in October was assumed to have the same depth to permafrost as the fully thawed active layer in August. An iterative model inversion scheme was used to estimate multiple active layer parameters by minimizing differences between the observed radar backscatter measurements and radar scattering model simulations. Initial sensitivity tests showed the capability of the model inversion in resolving subsurface active layer properties including surface freeze-thaw status, ALT and soil moisture in relation to independent in situ measurements from CALM sites.

## 2.3 Model sensitivity analysis

### 2.3.1 Regional sensitivity analysis

SOC fraction, soil moisture and snow cover conditions are among the most important factors controlling permafrost active layer conditions at the landscape scale (Lawrence and Slater, 2008; Jafarov et al., 2012; Zhang et al., 2014; Yi et al., 2015). Therefore, model sensitivity analyses were conducted to investigate the ALT sensitivity to uncertainties in regional SOC fraction, soil moisture and snow density (Figure 1). The modelled ALT uncertainties were calculated as the standard deviation between the model baseline simulations and a set of model sensitivity runs conducted over the study period by adding uncertainties into the regional SOC map (including total SOC content and vertical distribution), SM, and snow density data used as model inputs.

For the SOC fraction, we accounted for uncertainties associated with the total SOC content and vertical distribution within the top 1-m soil profile due to substantial uncertainties in these properties from available soil inventory records for the Arctic region (Ping et al., 2008; Burnham and Sletten, 2010). An uncertainty range of ±5 kg C m$^{-2}$ was assigned to the baseline SOC value from the ancillary SOC inventory data, based on reported uncertainties and comparisons with other Alaskan SOC estimates (Mishra et al., 2016). For each total SOC scenario, i.e. high, baseline, and low SOC scenarios, we performed three simulations to account for the uncertainties in the SOC vertical distribution: "surface", baseline, and "even" allocation scenarios, with a lower SOC density within surface soils in the baseline and "even" allocation scenarios (Figure 1). The total SOC content was assumed to decrease exponentially with depth along the soil profile (Jobbagy and Jackson, 2000; Meersmans et al., 2009; Hossain et al, 2015); two parameters, including the SOC density at the surface and a vertical decay parameter (k), were used to determine the soil carbon density for each model soil layer (Meersmans et al., 2009):

$$SOCC(z) = SOC0 \cdot \exp(-k \cdot z) \tag{1}$$

Where SOCC is the estimated soil organic carbon density (kg C m$^{-3}$) at a given soil depth, $z$ (cm); $SOC0$ and $k$ represent the surface SOC density and vertical decay rate (m$^{-1}$) with increasing soil depth, respectively. The k values were determined based on the reported SOCC profile for different biome types (Jobbagy and Jackson, 2000; Meersmans et al., 2009; Hossain et al., 2015). Boreal forest is characterized as having generally greater surface SOC accumulation than tundra for relatively undisturbed conditions (Hossain et al., 2015) and was thus assigned a larger k value (Figure S4). The prescribed k values for the three SOC vertical distribution scenarios range from 0.03 to 0.05 m$^{-1}$ for boreal forest and from 0.01 to 0.03 m$^{-1}$ for tundra and other vegetation biomes (Table S2). The soil carbon or organic fraction for each soil layer was estimated as:

$$f_{sc,i} = SOCC(z_i)/SOCC_{max} \tag{2}$$

Where $SOCC(z_i)$ is the estimated soil carbon density at the centre depth ($z_i$) of soil layer i and $SOCC_{max}$ =130 kg m$^{-3}$ is the maximum soil carbon density of peat soils (Farouki, 1981). Mineral soils may also contain a high soil carbon density but low soil organic fraction due to much higher bulk density. Therefore, the soil carbon fraction was adjusted based on an empirical relationship between soil carbon concentration and bulk density (Hossain et al., 2015) when the SOCC is below 40 kg C m$^{-3}$.

There are large uncertainties associated with soil moisture and snow cover parameters derived from satellite observations and reanalysis data. Initial validation of the SMAP NatureRun and L4SM soil moisture products indicated an un-biased RMSE below 0.05 $m^3$ $m^{-3}$, though this was primarily assessed for mineral soil type conditions within the continental USA (Reichle et al., 2017). For the model sensitivity analysis, a soil wetness uncertainty (±10%) was assigned to the SMAP NatureRun and L4SM soil moisture records based on prior global soil moisture assessments using MERRA reanalysis data (Yi et al., 2011). The ±10% wetness uncertainty translates into an uncertainty of ~ 0.04 $m^3$ $m^{-3}$ for mineral soils and ~ 0.08 $m^3$ $m^{-3}$ for organic soils which typically have a higher porosity. An uncertainty level of ±25% was assigned to the MERRA-2 snow density estimates based on comparisons with snow density observations derived from GPS (Global Positioning System) L-band backscatter signals from six Plate Boundary Observatory (PBO) sites across Alaska (Figure S5). However, the uncertainty in snow density was limited to ±20 kg $m^{-3}$ during the initial snow accumulation period, with snow density generally ranging from 100 to 200 kg $m^{-3}$. Compared with snow density, snow depth shows much larger temporal variability (Sturm et al., 2010), which makes it difficult to assign temporally varying uncertainty levels for the snow depth estimates. However, the above scheme partially accounts for uncertainties in the snow depth data due to a positive correlation between snow depth and density at longer time scales (McCreight and Small, 2014).

For both model baseline simulations and sensitivity runs, the model was spun-up for 50 years to bring the top 10-m soil temperature profile into dynamic equilibrium with model inputs for the year 2000, followed by a transit run from 2001 to 2015. Different model spin-up schemes may have a large impact on the model simulations; therefore, an additional initialization scheme was tested for the baseline model simulation. Because there were no data available from MODIS and SMAP NatureRun records prior to year 2000, the model was first initialized using MERRA-2 surface meteorology including air temperature, SM and snow data from 1980 to 1999, followed by a model transit run from 2000 to 2015 using the MODIS LST, SMAP SM and MERRA-2 snow data. The MODIS LST and MERRA-2 surface air temperature records showed overall consistent regional mean temperatures during the overlapping period. The two model spin-up schemes produced very similar regional ALT estimates for the year 2000 initial conditions; therefore, only the model simulations and results based on the first spin-up scheme were presented.

### 2.3.2 Modeled ALT sensitivity to landscape heterogeneity within the airborne flight transect

An integrated analysis of in situ ground measurements, airborne radar retrievals and soil process model simulations was conducted to verify modeled ALT simulations in relation to other observations and investigate the ALT sensitivity to spatial variability in soil organic carbon fraction and soil moisture. We selected four in situ CALM sites located within the airborne radar DH sub-region acquired in October 2015 for ALT comparisons with model simulations and radar retrievals (Table 2). Additional CALM sites are located within the DH sub-region (Figure S3); these sites are generally located near the validation sites but had very different landscape properties (including SOC fraction and soil saturation degree) from the

model inputs and radar retrievals, and were therefore not selected for the model comparisons. In particular, the Sagwon Hills MAT site was excluded due to potentially large uncertainties in the radar ALT retrievals ($21.0 \pm 0.07$ cm) due to reduced radar penetration and ALT sensitivity under very wet conditions (radar SM retrievals: 0.47 m$^3$ m$^{-3}$) as indicated by a significant negative correlation (R = -0.47, p < 0.1) between the radar ALT and SM retrievals within the ~1 km grid cell (18 × 18 pixels).

A model sensitivity analysis was conducted within the DH sub-region, which covers the area between 69.5°N and 70°N. This region was selected on the basis of having relatively higher radar SM and ALT retrieval accuracy (Figure S3). Above 70°N, the radar retrievals indicate very low SM levels (< 0.2 m$^3$ m$^{-3}$), likely due to active layer freezing in October. Below 69.5°N, the very wet soil conditions may introduce larger uncertainties in the radar ALT retrievals as discussed above. For the model sensitivity analysis, we first calibrated the soil porosity and active layer soil saturation degree over the DH sub-region by minimizing root mean square error (RMSE) differences between the spatially aggregated radar ALT retrievals and model ALT simulations derived using regionally averaged SOC, LST, and snow properties inputs. We then compared the spatial distributions of the 1-km aggregated airborne radar ALT retrievals and the 1-km model simulations to determine the ALT sensitivity to relatively coarse regional drivers including LST, soil wetness (% volumetric) and SOC fraction. Three model sensitivity runs were performed (Table 3). The model was first driven using 1-km MODIS LST inputs, but with regionally averaged SOC, snow and soil wetness conditions (Run1). The model was then driven using both 1-km MODIS LST and soil wetness derived using the above soil porosity estimate and radar-retrieved volumetric soil moisture, with regionally averaged SOC and snow conditions (Run2). The SOC map (Mishra et al., 2016) indicates high SOC levels (mean = 45 kg C m$^{-2}$) with low spatial variability ranging from 40 to 50 kg C m$^{-2}$ in this area. However, the soil inventory record may not adequately account for fine-scale variability in the SOC content that could result from local soil wetness variability (Mishra and Riley, 2015). Therefore, an additional model simulation (Run3) was conducted with similar drivers as Run1, but with a larger range of variability in the SOC fraction. Specifically, the Run3 scenario assumes the regional SOC distribution follows the statistical distribution of radar retrieved soil moisture across the DH sub-region from 69.5-70°N (Figure S3 c), resulting in an estimated SOC range from 21 to 69 kg C m$^{-2}$, and a mean value of 45 kg C m$^{-2}$. This statistical distribution was similar to the OLT distribution observed from field sampling data in northern Canada (Zhang et al., 2014).

## 3. Results

### 3.1 Regional ALT validation

#### 3.1.1 Comparison with in situ measurements

The model simulated soil temperatures at the selected tower sites using local site meteorology and prescribed SOC fractions based on in situ OLT data showed favourable performance in relation to the in situ measurements, with mean R values above

0.90 and mean RMSE values less than 2.5 °C (Table S3; Figure S6-S7). The model simulated maximum soil thaw depth (i.e. ALT) was within the ALT uncertainty range from the in situ data (Figure 2). At the boreal forest site, the model simulated ALT ($81 \pm 15$ cm) during the study period (2001-2015) was much larger than the ALT value reported at the tower site (~ 43 cm, Nakai et al., 2013); however, the model simulated ALT was close to the ALT ($74 \pm 17$ cm) calculated from the in situ soil temperature measurements during the observation period (2011-2013). The seasonality of the model simulated soil thaw depth also generally followed the pattern of soil thaw depth calculated from the in situ soil temperature observations (Figure S8). However, limited deep soil temperature measurements at the site (only available at 40 cm and 100 cm) may contribute significant uncertainty to the calculated soil thawing depth. At the two tundra sites, the model simulated ALT generally falls within the range of ALT values reported at the tower sites. At the AK-Imn site, the model simulated mean ALT ($47 \pm 8$ cm) was slightly shallower than the observations ($53 \pm 5$ cm) at the CALM site, and the IMNAVAIT 1-km grid cell encompassing the tower site. At the US-Atq site, the model simulated mean ALT ($37 \pm 9$ cm) was close to the in situ ALT (~ 40 cm) reported by Oechel et al. (2014). The model simulations at the two tundra sites showed overall later soil thaw onset in spring and earlier autumn soil freeze onset than the boreal forest site, resulting in a shallower ALT.

The model simulated mean ALT generally increased with decreasing latitude and permafrost probability (PP) indicated by a satellite and soil inventory based PP map (Pastick et al., 2015, Figure 3b), with relatively shallow ALT values in areas with higher PP, including the Alaska North Slope and Seward Peninsula, and deeper ALT values in sporadic and isolated permafrost areas (i.e. PP < 50%) including most of the Alaska interior and southwestern region (Figure 3). The model showed better performance against in situ ALT measurements from CALM sites with higher PP. Sites without a consistent presence of permafrost within 3-m surface soils during the study period were excluded from the comparisons and were mostly distributed in areas with PP < 50%. A total of 51 CALM sites meeting the validation criteria were used for the model comparisons, while 33 of these sites were located in areas with PP ≥ 70%. The modelled ALT was generally deeper than the ALT observations for sites located in areas with PP < 70%. The modelled ALT showed relatively low correspondence with the in situ measurements when all 51 sites were included (R = 0.46; mean bias = 17.39 cm; RMSE = 40.51 cm), but with better agreement for sites located in areas with PP ≥ 70% (R = 0.60; mean bias = 1.58 cm; RMSE = 20.32 cm). Larger differences between model simulations and in situ ALT measurements in areas with lower permafrost probability is not unexpected due to strong surface heterogeneity in permafrost conditions, leading to larger discrepancy between model simulations representing a single ALT value for each 1-km$^2$ grid cell and the point-scale measurements. In those areas, the satellite and soil inventory based PP map indicated permafrost occurrence within 1-m surface soils well below 100%, while the in situ measurements showed ALT generally shallower than 1m (Fig. 3d).

### 3.1.2 Integrated analysis of radar retrievals and model simulations

The modelled ALT results were similar to the in situ ALT measurements and airborne radar retrievals within the DH sub-region (Figure 4). The DH sub-region is located within the northern Alaska continuous permafrost zone (PP ≥ 90%). The

model simulations, radar retrievals and in situ measurements all showed the lowest ALT values (< 40 cm) at the most northern site (West Dock) within the DH sub-region, but with larger differences at the other DH sites. The model simulations were very close to the radar ALT retrievals at the Deadhorse and Franklin Bluff sites, and similar to the in situ observations at the Sagwon Hills site, though the radar ALT retrievals indicated shallower ALT conditions than both model results and observations at this site. The radar retrievals likely underestimated ALT for the Sagwon Hills area due to very wet soil conditions observed at this site (SM > 0.4 $m^3$ $m^{-3}$, Figure S3), which reduced microwave penetration depth and active layer sensitivity. The soil moisture impact on the radar ALT retrievals is indicated by a significant negative correlation (R < -0.45, p < 0.1) between the radar ALT and SM retrievals at both Sagwon Hills sites. Relatively large differences were observed between the modelled ALT values and in situ observations at the Deadhorse and Franklin Bluff sites, though the model results were similar to the radar ALT retrievals at these sites. Despite these differences, the modeled ALT showed overall consistent inter-annual variability (R > 0.5, p < 0.1) for all of the DH sub-region sites except West Dock, which had a deeper organic layer and smaller ALT inter-annual variability than the other sites (Figure S9 and Table 2).

The regional model sensitivity analyses for the DH sub-region between 69.5°N and 70°N indicates the important role of the SOC fraction on the model simulated ALT pattern (Figure 4c and Table 3). The DH sub-region was selected for the model sensitivity analysis due to lower uncertainties in the airborne radar ALT and SM retrievals as discussed in Section 2.3.2. Using the spatial average of the radar ALT retrievals (39.59 ± 0.06 cm), the soil model estimated a mean soil porosity of 0.61 $m^3$ $m^{-3}$, and mean soil wetness of 63% for the active layer, which was close to the soil wetness estimates for the same area derived from the SMAP L4SM product (62% - 66%). The soil model simulations derived using the 1-km MODIS LST inputs and regional mean SOC and SM inputs (Run1) showed a slightly smaller mean ALT of 37.90 ± 0.04 cm. Model simulations derived using the 1-km MODIS LST and airborne radar SM retrievals as inputs (Run2) showed a similar ALT distribution as the Run1 results, but with larger spread (40.36 ± 0.11 cm). The soil model simulations based on similar inputs as Run1, but accounting for the statistical distribution of the regional SOC inputs (Run3) resulted in a more consistent ALT spatial distribution with the radar ALT retrievals (mean ALT = 41.61 ± 0.07 cm). The effect of snow cover heterogeneity on the ALT distribution was not investigated here due to the coarse resolution of the MERRA-2 snow data (~ 0.5°) and thus small differences in the interpolated 1-km snow depth and density data within the sub-region.

### 3.2 Regional ALT sensitivity to environmental variables

The model results indicated widespread ALT deepening during the 2001-2015 study period, with 79.2% of simulated permafrost (ALT < 300 cm) areas showing positive trends (Figure 5). However, only ~ 24.0% of estimated permafrost areas showed significant (p < 0.1) positive ALT trends due to large interannual variability in model simulated ALT and relatively short (15-year) data record. Very few areas (< 0.3% of the domain) showed significant negative ALT trends. The model simulations showed relatively smaller ALT trends (0.32 ± 1.18 cm $yr^{-1}$) in continuous permafrost areas of northern Alaska, which has a generally colder polar climate and more stable permafrost conditions. The model results indicated much larger

positive ALT trends ($> 3$ cm yr$^{-1}$) across central and southern Alaska, which is characterized by warmer climate conditions and more sporadic permafrost conditions (PP < 50%). Both modelled ALT and associated temporal trends generally increase with decreasing permafrost probability (Figure 3 and Figure 5). Relatively large spatial variability in the estimated ALT trend also occurs in areas with lower permafrost probability.

The ALT trends and spatial variations are mainly affected by the accumulated thawing degree days during the snow-free period (R = 0.60 ± 0.32, Figure 5b and Table S4). Model simulated ALT during the study period was significantly correlated with MODIS LST thawing degree days during the snow-free period, with regional mean correlations above 0.51 (p < 0.1) for areas with PP ≥ 20%. The MODIS LST record indicates a strong warming trend in spring (0.095 ± 0.09 °C yr$^{-1}$) and a non-significant warming trend in summer (0.006 ± 0.066 °C yr$^{-1}$), which leads to a longer snow-free season and associated increase in the heat input to the soil. The warming trend is commensurate with a positive trend in MODIS LST thawing degree days during the snow-free season (0.415 ± 0.982 °C yr$^{-1}$). Both the MODIS snow cover product and the MERRA-2 snow depth data show significant lengthening of the snow-free season in central and southwestern Alaska (Figure 6), mainly due to earlier snow offset in spring. The autumn snow onset trend is more variable across the region, and an overall earlier snow onset in northern Alaska mainly contributes to a shorter snow season in those areas. A reduced correlation between MODIS LST thawing degree days and model simulated ALT in areas with PP < 20% is likely caused by larger uncertainties in the model simulations in these areas as discussed below.

## 3.3 Uncertainties in regional ALT simulations

The model sensitivity analysis indicated significant uncertainties influencing estimated ALT patterns and trends from several sources (Figure 7). The model simulated ALT is associated with large uncertainties in areas with lower SOC fraction (particularly for surface conditions) and lower permafrost probability. Uncertainties in the model simulated ALT due to uncertainties in the total SOC content increases from a few centimetres (~ 5%) in continuous permafrost areas to approximately 50 cm (~ 45%) in sporadic permafrost areas. ALT uncertainties due to the soil carbon vertical distribution show a similar pattern, but with slightly lower magnitude. In areas where PP < 70%, the model simulated mean ALT increased by 26% and the loss of model simulated permafrost areas with ALT < 300 cm doubled with reduced total SOC content (Table S5 and Figure 8). In comparison, model simulated areas with ALT < 300 cm in areas where PP ≥ 70% showed negligible response to SOC variability due to predominantly shallower ALT in these areas. Here, the ALT < 300 cm threshold is used to define the boundary of model estimated near surface permafrost extent over the Alaska domain. Larger variability in the model simulated mean ALT and accelerated permafrost loss in areas defined by PP < 70% were also observed when less SOC was allocated in the surface soils (e.g. "even" allocation scenario). The inverse relationship between surface SOC fraction and ALT in the model reflects the strong insulating effect of surface organic soils.

The results indicated large model uncertainty associated with the representation of snow cover conditions, particularly the low snow density scenario (Table S6 and Figure 7b). For the low snow density scenario, the model simulated ALT increased by 56% from 61 cm (baseline) in the more continuous permafrost zone (PP $\geq$ 70%), and by 49% from 146 cm in areas with PP < 70%, while the model simulated loss of areas with ALT < 300 cm in the permafrost zone (PP < 70%) from 2001 to 2015 doubled compared with the baseline simulation. The model results also showed significant (p < 0.01) loss of areas with ALT < 300 cm even in areas with PP $\geq$ 70% for the low snow density scenario. However, the model may overestimate uncertainties associated with the low snow density scenario. The MERRA-2 snow density generally ranges from 200 to 250 kg m$^{-3}$ during the snow season, which is near the lower range of previous estimates especially for maritime and tundra snow cover (Sturm et al., 2010; Bormann et al., 2013). The MERRA-2 snow density did not show a significant low bias compared with the PBO site observations; however, the MERRA-2 snow depth data generally showed positive bias compared with the PBO snow depth data (not shown), which may lead to an overestimation of MERRA-2 snow density. Therefore, the model simulations from the low snow density scenario may significantly overestimate snow insulation effects and ALT uncertainty, especially in southwestern Alaska with more variable snow cover conditions.

The uncertainty contributed from the SMAP SM data to modelled ALT is relatively small compared with SOC distribution and snow density contributions (Figure 7b). The model simulations for the "high SM" scenario promoted generally deeper ALT levels and slightly larger loss of permafrost areas (ALT < 300 cm) in the permafrost zone (PP < 70 %) than the baseline simulations due to enhanced effects of SM on soil heat transfer and heat storage (Table S5). The ALT sensitivity to SM showed limited variability under different SOC levels (Figure S10). However, the accuracy of SMAP SM data in boreal and tundra ecosystems requires further investigation. In addition, the SMAP SM data did not account for SM redistribution associated with permafrost degradation during the study period, which may have a significant impact on soil heat transfer especially in discontinuous and sporadic permafrost areas.

## 4. Discussion

Our model estimates of regional permafrost active layer conditions over Alaska are generally consistent with previous studies. A study using an empirical data fusion and modelling approach incorporating extensive field observations and spatial environmental datasets (Pastick et al., 2015) estimated that near-surface (< 100 cm) permafrost encompasses 38% of mainland Alaska, with a mean ALT of 50 cm. Our model baseline simulations indicate a similar near-surface (< 100 cm) permafrost extent encompassing ~ 40% of the Alaska domain, with a mean ALT of 58 cm. Another study using spatially referenced soil profile data and environmental variables produced ALT estimates across Alaska ranging from 14 to 93 cm, with a spatial average of 46 cm (Mishra and Riley, 2014). A follow-on study estimated the mean ALT across Alaska to be between 42 cm and 49 cm with 95% confidence (Mishra et al., 2016). Both studies indicate a dominant existence of near-surface permafrost across the Alaskan domain, which is larger than our model results and the previous study by Pastick et al.

(2015). Our model predicts relatively stable permafrost conditions in continuous permafrost areas during the study period, which is consistent with previous reports (Osterkamp, 2007; Jafarov et al., 2012; Nicolsky et al., 2017). Our estimate of the ALT trend in those areas ($0.32 \pm 1.18$ cm yr$^{-1}$) is also comparable with a regional modelling experiment in Northern Alaska (Nicolsky et al., 2017).

Our model results indicate widespread active layer deepening in the study domain from 2001 to 2015, with generally larger positive trends (> 3 cm yr$^{-1}$) in discontinuous and sporadic permafrost areas including central and southern Alaska, and smaller trends (~ 0.32 cm yr$^{-1}$) over colder and more continuous permafrost areas of northern Alaska (Figure 5). Our analysis indicates that a longer snow-free period and concurrent surface warming are mainly responsible for ALT deepening during the study period. Previous studies have also noted that the ALT is primarily determined by the cumulative thermal history of the ground surface during the summer thaw season (Zhang et al., 2005; Osterkamp, 2007). A few studies based on satellite observations and modelling indicate that regional warming and a longer thaw season have led to widespread permafrost degradation and active layer deepening in permafrost areas (Yi et al., 2015; Park et al., 2016). Alaska shows a strong spring warming trend during the study period, which results in significantly earlier snow melt and a longer snow free season. Early snow melting in the spring increases energy inputs into soils and generally enhances soil warming, which may promote ALT deepening and permafrost degradation due to the snow cover-climate feedback (Lawrence and Slater, 2010). On the other hand, the snow onset shows more variable trends across the region, with northern Alaska generally showing an earlier snow onset trend. However, the relationship between autumn snow onset and soil warming is more variable depending on the timing of snowfall and local climate conditions (Yi et al., 2015). Early snow onset may enhance thermal buffering of cold surface temperatures, and promote soil warming in colder climate zones (Zhang, 2005).

Our results indicated large uncertainties in model estimated ALT associated with uncertainties in both the spatial variability and vertical distribution of SOC. Soil organic matter is a key factor affecting permafrost active layer processes due to its effects on soil thermal and hydraulic properties (Lawrence and Slater, 2008). There are substantial differences among available SOC datasets in northern permafrost areas, partially due to insufficient field data sampling and strong SOC variability associated with local vegetation, terrain, disturbance and soil moisture heterogeneity (Ping et al., 2008; Johnson et al., 2011). A relatively fine-resolution (~ 50 m) SOC dataset generated using more than 500 soil profiles in Alaska was used to parametrize the model SOC distributions. However, the SOC dataset may still underestimate SOC variability associated with large heterogeneity characteristic in boreal and arctic landscapes. An integrated analysis of the airborne radar retrievals and soil process model sensitivity runs over the DH sub-region (Figure 4) showed that the model can better simulate ALT spatial heterogeneity after introducing a statistical distribution of the regional SOC spatial pattern. The model sensitivity analysis also showed that uncertainty in the vertical SOC distribution contributes significantly to the model estimated ALT uncertainty (Figure 8) due to strong insulation effects of surface organic soils (Jafarov and Schaefer, 2016). The SOC content was assumed to decrease exponentially with increasing depth from the surface (Eq. 1), which may significantly

underestimate the SOC of deep soils in areas strongly affected by cryoturbation (Ping et al., 2008; Burnham and Sletton, 2010). However, this process should have a relatively limited effect on the estimated soil carbon fraction due to general increases in soil bulk density and thus lower soil carbon concentration with depth (Hossain et al., 2015). Better information on the spatial and vertical distribution of SOC stocks would provide the single largest improvement in ALT accuracy,

enabling more accurate predictions of permafrost active layer processes and climate feedbacks in regional and global carbon and climate models (Zhang et al., 2014; Mishra and Riley, 2015).

The effects of soil organic matter on ground temperature evolution are also influenced by soil moisture content, which affects soil thermal conductivity and heat exchange processes (Hinkel and Nelson, 2003; Nicolsky et al., 2017). Our study

may underestimate the modeled ALT uncertainties associated with SMAP SM data. Large uncertainty is associated with global reanalysis or satellite SM data (Yi et al., 2011). This uncertainty is due to many factors, including insufficient understanding of the effect of permafrost-thaw induced transitions on active layer hydrology (Rawlins et al., 2013), and the predominance of wet soil conditions and standing water in permafrost landscapes, which constrains satellite microwave penetration and sensitivity to active layer properties (Du et al., 2015). The SMAP SM data did not account for soil drainage

and soil moisture redistribution with permafrost thaw and ALT deepening (Walvoord et al., 2016), which may result in overestimation of ALT trends in areas with deep active layers and wet soil conditions, characteristic of much of western Alaska (Figure 5). Our initial model sensitivity analysis over the DH sub-region did not show significant improvement in ALT results using fine-resolution (~ 50 m) airborne radar SM retrievals. The lack of model improvement may be due to uncertainty in the dielectric conversion model parameters used for the radar SM retrieval, but may also indicate a need for

better parameterization of soil moisture effects on model soil heat transfer processes. A close association between SOC and local topographic attributes, including soil wetness, has been reported (Mishra and Riley, 2015); this may explain why model simulations derived using a statistical SOC distribution following the radar SM pattern produced better ALT performance relative to the radar retrievals. Other potential geophysical retrievals from multi-frequency radar remote sensing, including SOC, freeze-thaw and SM profile (Du et al., 2015; Bartsch et al., 2016), may enable improved model representation of

processes affecting permafrost active layer conditions.

Other uncertainties in the model inputs and structure may also result in large uncertainties in our regional ALT estimates. Large-scale satellite observations and global reanalysis data are unable to resolve finer scale microclimate variations influencing the ground thermal regime, including spatially complex snow cover properties influenced by local topography,

vegetation and winds (Liston and Sturm, 1998; Gisnas et al., 2016). These effects may be more pronounced over more complex terrain, including southwest Alaska, where the model shows larger uncertainties in ALT simulations and trends (Figure 5). The model uses satellite skin temperature (i.e. MODIS LST) to define the upper boundary conditions, which does not account for vegetation canopy effects on ground thermal conditions, and may add significant uncertainties in dense vegetation areas. Increasing disturbance from thermokarst and wildfire alter microclimate and SM conditions, vegetation

cover and SOC stocks, triggering a series of physical and ecological changes, all closely related to the dynamics of ground-ice evolution and permafrost degradation (Jorgenson et al., 2006; Osterkamp et al., 2009; Grosse et al., 2011). These effects are not adequately represented by the current model. Additional airborne radar sampling targeting regional disturbance gradients may provide the necessary information for representing these processes in the regional modelling framework.

**5. Conclusions**

We developed a satellite-based modelling framework for permafrost active layer mapping at landscape scale (~ 1 km) and applied it to the Alaskan domain. Local scale (~ 50 m resolution) maps of ALT and SM derived from combined low frequency (L+P-band) airborne radar remote sensing were used with in situ ground measurements to evaluate the model simulations. The model estimated ALT was more similar to in situ observations and airborne radar retrievals in more

continuous permafrost areas (PP ≥ 70%) than in lower permafrost probability areas. The model simulations indicated widespread active layer deepening since 2001, with larger positive trends in discontinuous and sporadic permafrost areas over central and southern Alaska, and generally smaller trends in colder and more stable permafrost areas of northern Alaska. The ALT deepening is mainly driven by surface warming and regional trends toward a longer snow-free season. Areas with lower SOC fraction, especially in surface soil layers, showed larger ALT uncertainties and stronger sensitivity to

regional warming trends. A spatially integrated analysis of the airborne longwave (P+L-band) radar retrievals and model simulations confirmed the important role of SOC spatial variability and vertical profiles in affecting ALT accuracy. Additional AirMOSS/UAVSAR radar measurements will become available from the ABoVE airborne campaign in Alaska and western Canada, representing more extensive climate, terrain and vegetation conditions, and allowing for further testing and refinement of the modelling framework across a larger domain. Potential mapping of surface organic layer, freeze-thaw

and soil moisture profiles using the combined low frequency radar data may enable substantial improvements in the way coarser landscape models represent key processes and sub-grid spatial heterogeneity, enabling more accurate predictions of boreal and arctic environmental changes.

*Data availability*. The Alaskan ALT maps produced by this study are available at http://ntsg.umt.edu and will be archived
and distributed for public access through the NASA ABoVE archive at the NASA ORNL DAAC (https://daac.ornl.gov/). The radar ALT and SM retrievals are available upon request. Other data used in this study were obtained from free and open data repositories.

*Author contributions*. Y.Y. and J.S.K. initiated the study; Y.Y. did all calculations and wrote the paper, with inputs from
30 J.S.K.. R.C., M.M., R.H.R. contributed to the data and discussed the results. U.M, D.Z. and W.C.O. contributed to the data and provided feedbacks on the final version.

*Competing interests.* The authors declare no conflict of interest.

*Ackowledgements.* Funding for this study was provided by NASA (NNX14AO23G, NNX15AT74A). The authors thank T. Nakai and E. Euskirchen for providing the in situ tower data in Alaska.

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

**Table 1: Geospatial datasets used as primary inputs for the soil process model over Alaska.**

|  | Data source | Spatial resolution | Temporal period | Temporal resolution |
|---|---|---|---|---|
| Surface temperature | MOD11A2* | 1 km | 2000-present | 8-day |
| Snow depth/density | MERRA-2 | 0.5° | 2000-present | Daily |
| Snow cover extent | MOD10A2[†] | 500 m | 2000-present | 8-day |
| soil moisture | SMAP | 9 km | 2001-2015 (NatureRun) 2015-present (L4SM) | Daily |

* Wan et al., 2015; [†] Hall and Riggs, 2016

**Table 2: In situ CALM sites covered by the UAVSAR and AirMOSS airborne radar flights along the Dalton Highway (DH) in October 2015. The information on OLT and soil moisture conditions was obtained from in situ site measurements. The local scale (~ 50 m resolution) radar ALT retrievals were averaged within an 18 × 18 pixel window (~ 1 km × 1 km) to compare with the 1-km soil model ALT simulations. The correlations (R) between the in situ observations and model ALT estimates were calculated for the 2001-2015 study period.**

| Site name | Location | OLT (cm) | Soil moisture condition | ALT (cm) | | | R (in situ vs model) |
|---|---|---|---|---|---|---|---|
| | | | | In situ | Radar | model | |
| West Dock 1 ha grid | 70°22′N, 148°33′W | 34 | wet | 37.0 | 29.7 ± 0.13 | 33.0 | 0.27 |
| Deadhorse | 70°10′N, 148°28′W | 15 | wet | 73.0 | 44.2 ± 0.10 | 44.0 | 0.61* |
| Franklin Bluff | 69°41′N, 148°43′W | 23 | wet | 68.0 | 41.6 ± 0.11 | 44.0 | 0.53* |
| Sagwon Hills MNT | 69° 26′N, 148°40′W | 9 ± 1.2 | moist | 57.0 | 36.8 ± 0.08 | 53.0 | 0.53* |

* indicates $p < 0.05$

**Table 3: Model drivers for the three model sensitivity runs conducted within the Alaska DH sub-region. The model LST and SM inputs were derived from MODIS (MOD11A2) observations and airborne radar SM retrievals.**

|      | LST   | SM            | SOC                      |
|------|-------|---------------|--------------------------|
| Run1 | 1-km  | Regional mean | Regional mean            |
| Run2 | 1-km  | 1-km          | Regional mean            |
| Run3 | 1-km  | Regional mean | Statistical distribution* |

**\*** The statistical distribution of SOC followed the statistical distribution of 1-km radar SM retrievals for areas between 69.5°N and 70°N (**Figure S3 c**).

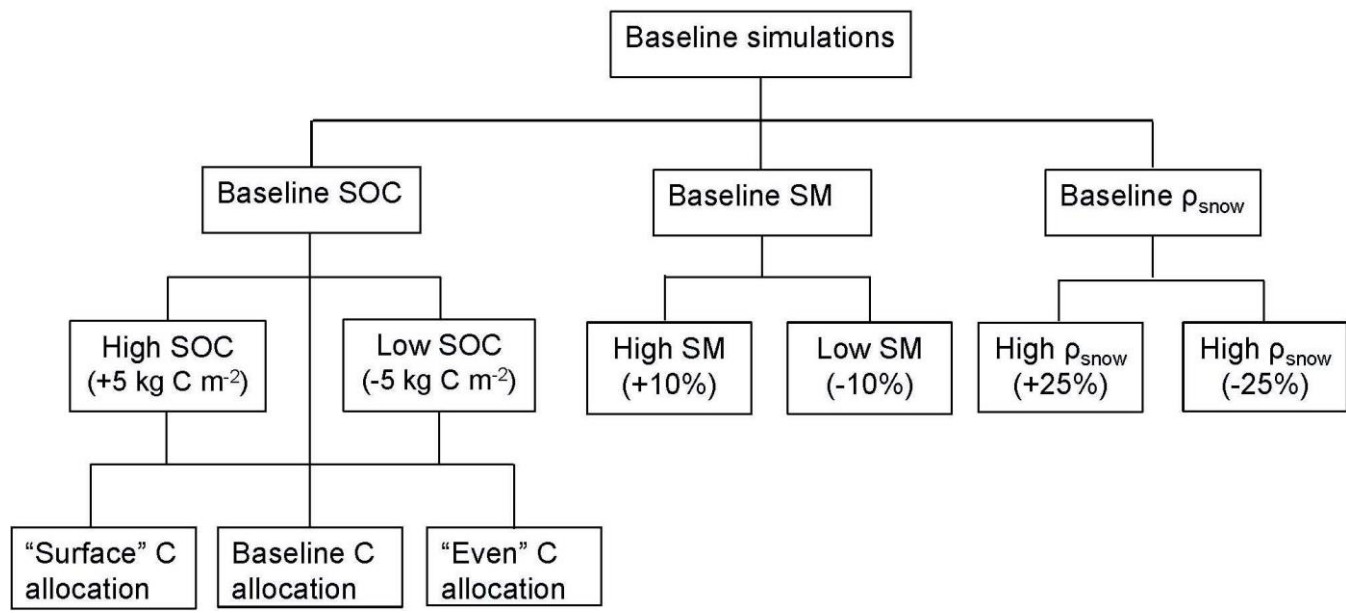

**Figure 1: Flow diagram describing the regional model sensitivity runs, accounting for uncertainties in total SOC content, SOC vertical distribution, SM and snow density ($\rho_{snow}$). Three SOC vertical allocation schemes were represented, including "surface", baseline and "even" allocation scenarios (Table S2).**

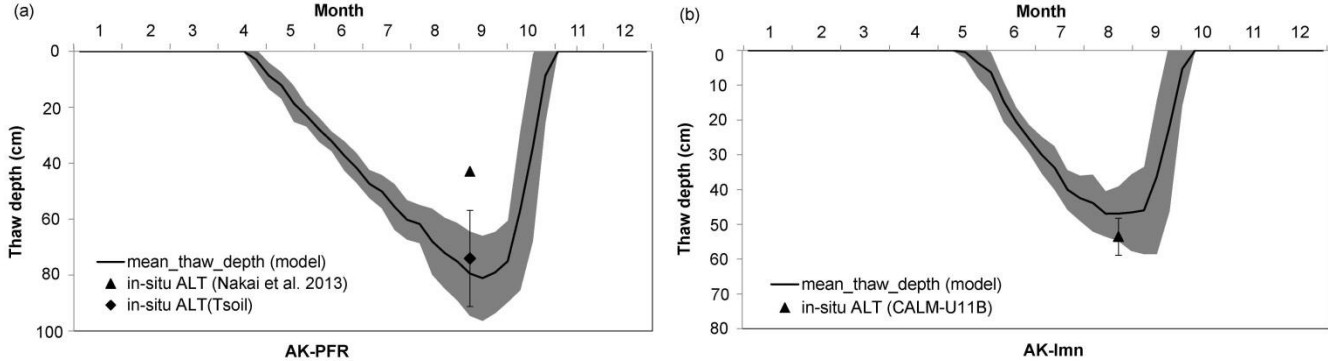

**Figure 2: Model simulated mean (2001-2015) soil thawing depth at the boreal forest (a: AK-PFR) and tundra (b: AK-Imn) sites, relative to in situ ALT values. At the AK-PFR site, the in situ ALT value reported at Nakai et al. (2013) was different from the ALT value calculated from in situ soil temperature (Tsoil) measurements, while at the tundra site, the in situ ALT was calculated from CALM site observations located within the IMNAVAIT 1-km model grid cell encompassing the AK-Imn tower site. Vertical error bars and dark gray shading indicate 1 standard deviation variability in soil thaw depth or ALT during the study period.**

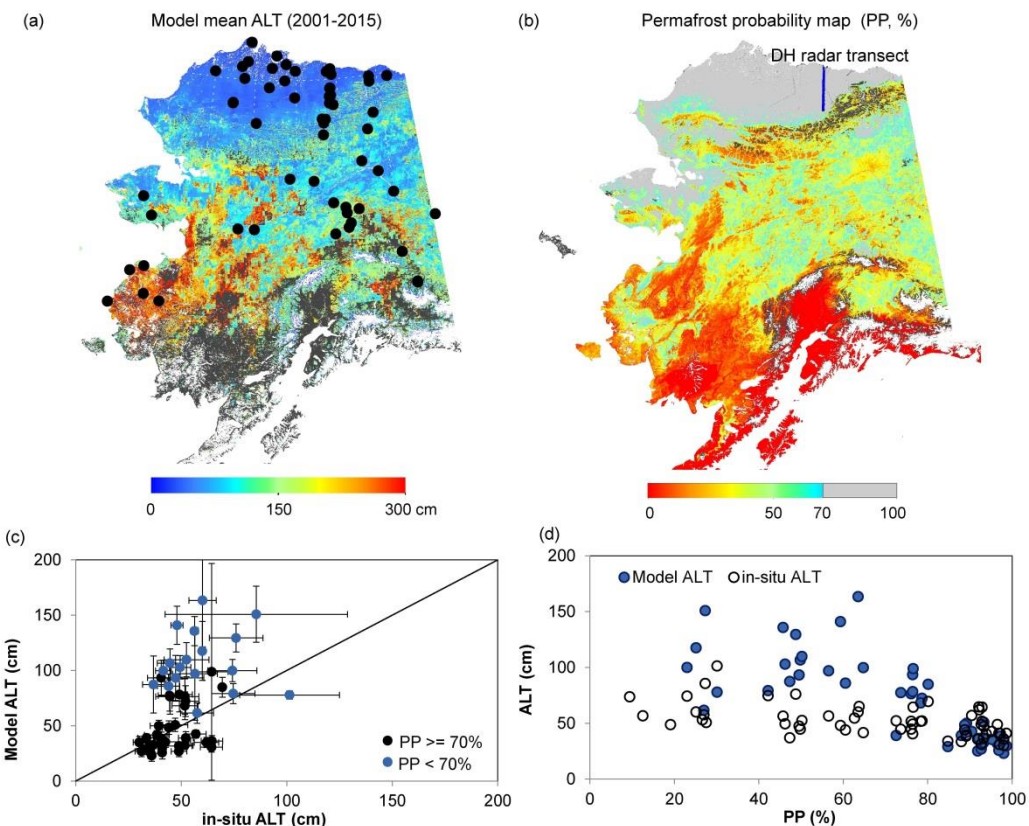

**Figure 3: Model simulated ALT and its performance against in situ CALM sites for different permafrost probability (PP) zones: (a) model simulated 1-km mean ALT map from 2001 to 2015; (b) a satellite-based permafrost probability map (Pastick et al., 2015); (c) comparisons of model simulated ALT against in situ CALM sites for different PP zones; (d) the changes of model and in situ observed ALT with permafrost probability. The areas with ALT greater than 300 cm depth are shown in dark gray (a). In panel (b), the areas with PP ≥ 70% are shown in gray, while areas outside of the PP classification are shown in black. The blue line in panel (b) indicates the location of the airborne DH radar flight transect used for model evaluation (Figure 4). The error bars in (c) represent the standard deviation of either model simulated or in situ observed ALT during the overlapping period.**

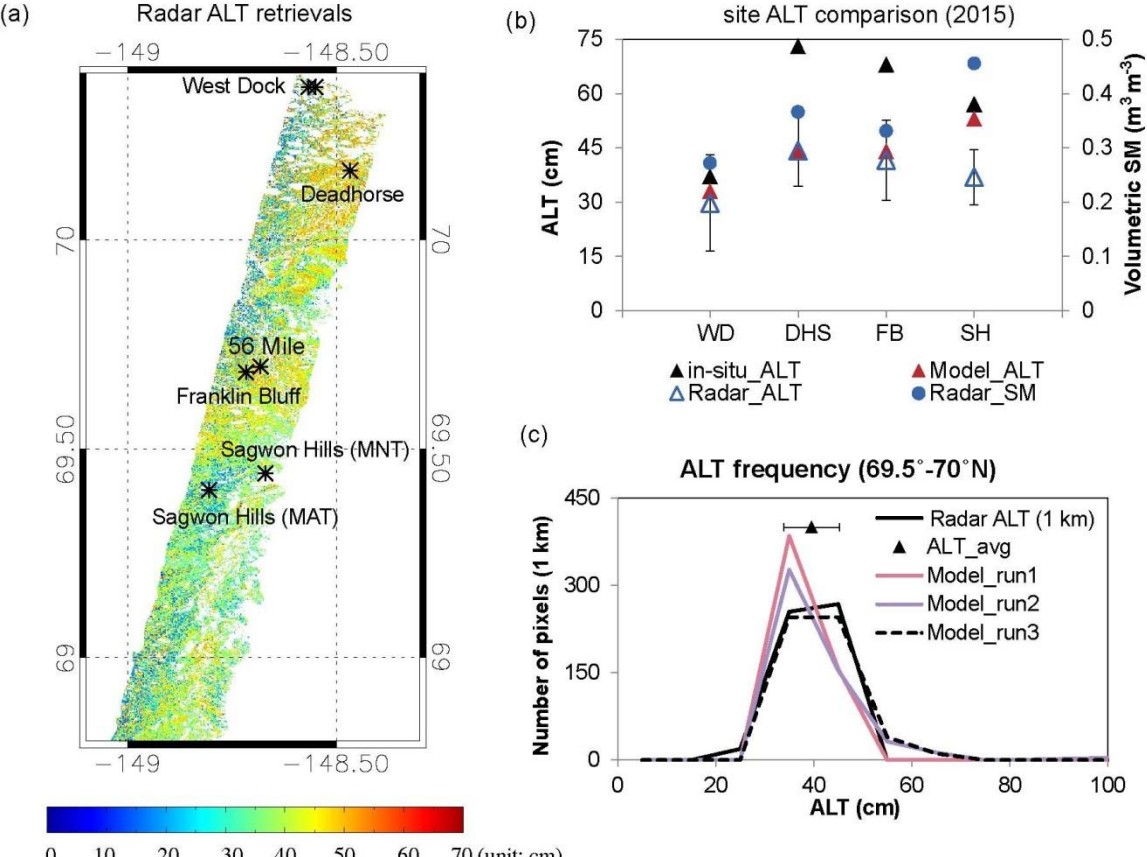

**Figure 4: Comparison of radar ALT retrievals with in situ CALM site observations and model simulations within the Alaska Dalton Highway (DH) sub-region defined from the airborne radar (AirMOSS, UAVSAR) flight transect in October, 2015: a) the radar retrieved ALT map from the combined (P+L-band) low frequency radar backscatter measurements, with areas indicated as open water, perennial ice/snow and developed areas masked out; b) comparisons of the in situ ALT observations, radar retrievals and model simulations at the CALM sites, including West Dock (WD), Deadhorse (DHS), Franklin Bluff (FB), and Sagwon Hills MNT (SH); c) comparisons of the ALT spatial distributions derived from the radar retrievals and model simulations for the DH latitudinal zone between 69.5 and 70°N. Different model runs were driven using different regional drivers (Table 3). ALT_avg is the regional mean of radar ALT retrievals with error bars representing the standard deviation.**

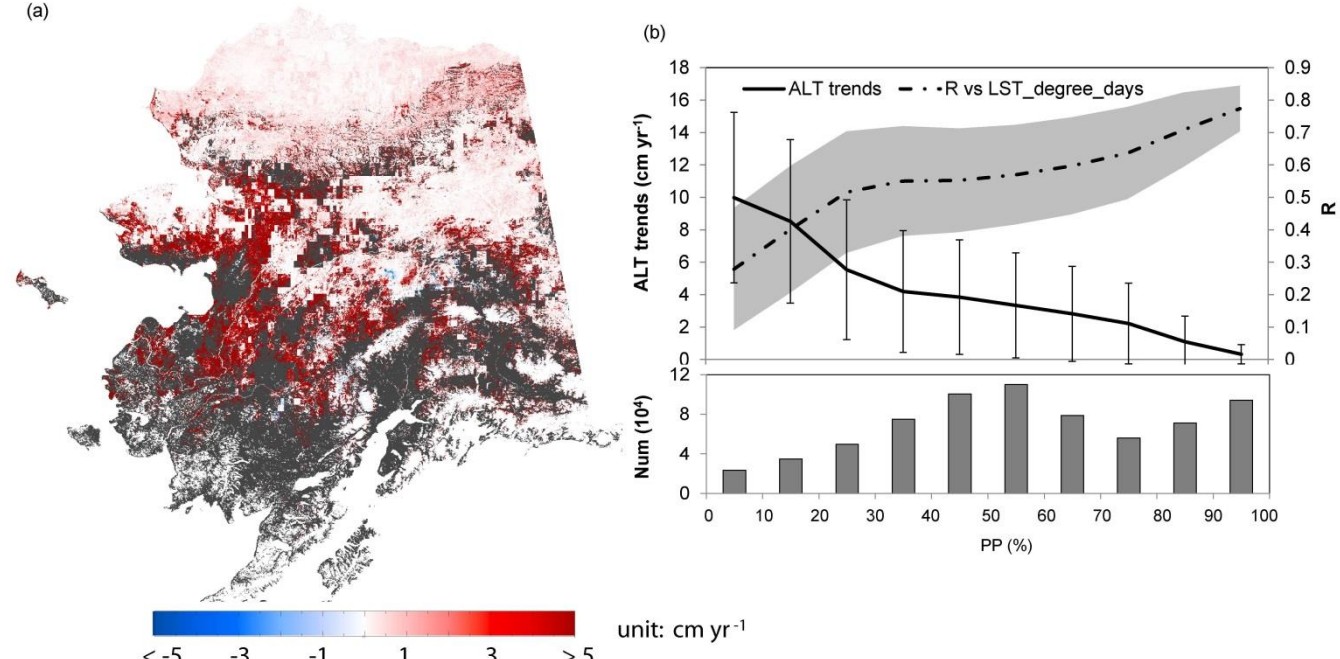

**Figure 5: Model simulated ALT trends and correlation with LST thawing degree days from 2001 to 2015: (a) simulated ALT trends over the Alaska domain, where areas with ALT > 3 m are shown in dark gray; (b) the distribution of ALT trends and correlations with MODIS LST degree days during the snow free season (upper panel) within different permafrost probability (PP) zones (Figure 3b); vertical error bars (dark gray) indicate 1 standard deviation for the regional ALT trend and correlation coefficient, respectively. The number of 1km pixels represented within each PP zone is shown in the lower panel (b).**

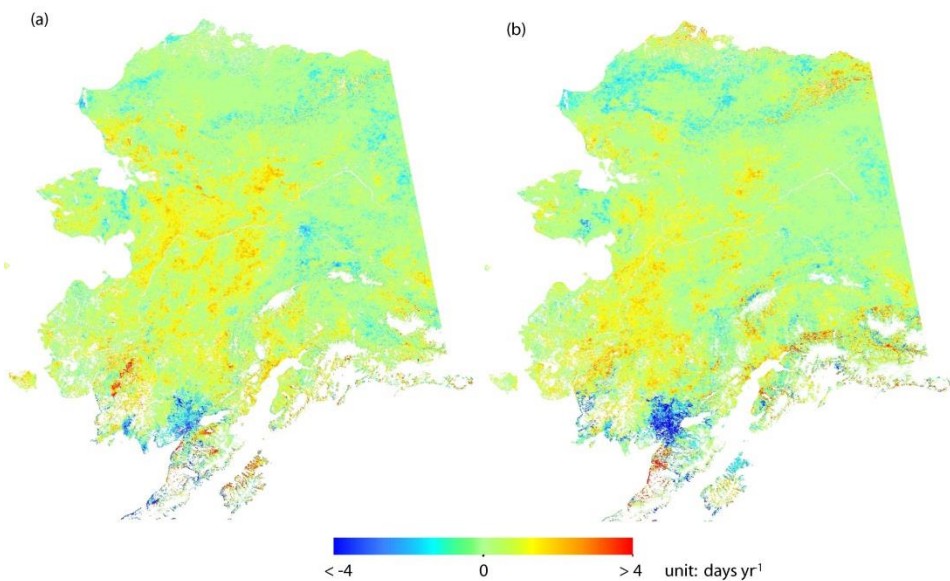

**Figure 6: Recent trends (days yr⁻¹, 2001-2015) in the annual snow-free period derived from the MODIS snow cover extent (SCE) product (MOD10A2, a) and the MERRA-2 snow depth data after filtering using the MOD10A2 SCE observations (b).**

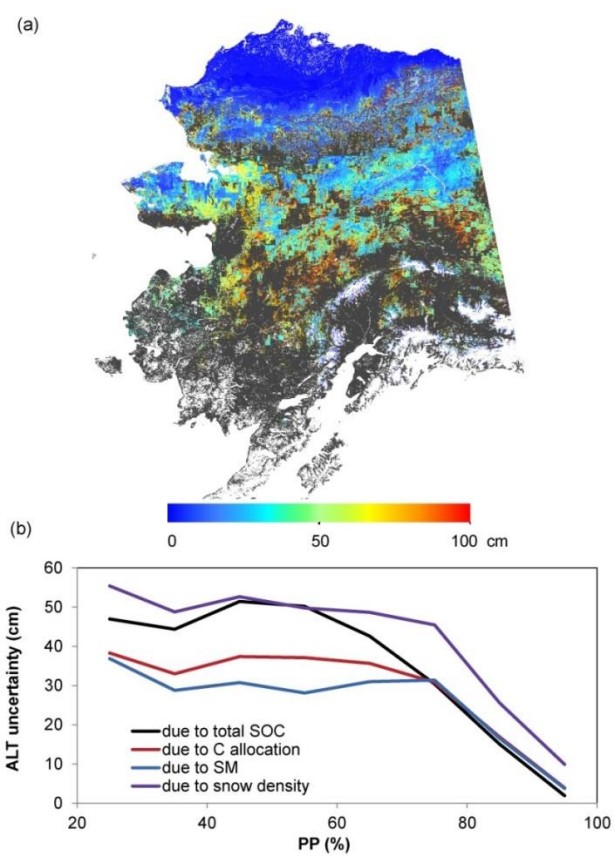

**Figure 7:** **Model estimated ALT uncertainty associated with uncertainties in SOC fraction, SM and snow cover properties: (a) spatial pattern of model simulated ALT uncertainty due to uncertainty in the total SOC content; the areas without consistent near-surface (< 300 cm) permafrost from all sensitivity runs were shown as dark gray. (b) shows the distribution of model mean ALT uncertainties associated with uncertainties in total SOC content, soil carbon allocation, soil moisture and snow density for different permafrost zones (Figure 3b). The ALT uncertainties were calculated as the standard deviation between the model baseline simulations and the sensitivity runs by adding uncertainties in the regional SOC map, soil moisture and snow density data (Figure 1).**

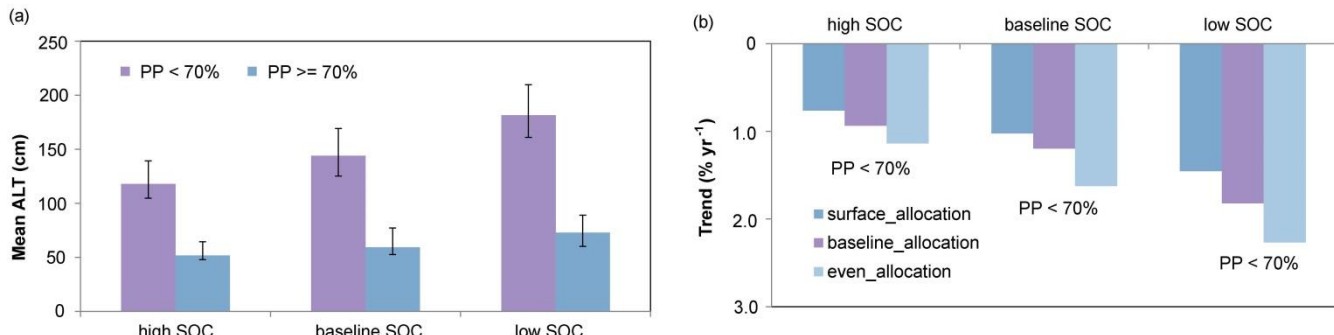

**Figure 8: The sensitivity of model simulated ALT to different SOC levels in two different permafrost probability (PP) zones (Figure 3b): (a) the model simulated mean ALT derived for different SOC levels, where error bars represent the variability in model simulated ALT due to different SOC allocation schemes (i.e. surface or even allocation); (b) shows the trends (% yr$^{-1}$, 2001-2015) in model simulated areas with ALT < 300 cm in the lower permafrost probability zone (PP < 70%) in proportion to model simulations in 2000.**

