# Peer review of "Characterizing permafrost active layer dynamics and sensitivity to landscape spatial heterogeneity in Alaska"

_The Cryosphere, 2017_

## Referee Comment (RC1) · Anonymous Referee #1 · 21 Jul 2017

Comments on "**Characterizing permafrost soil active layer dynamics and sensitivity to landscape spatial heterogeneity in Alaska**" by Yi et al., published in The Cryosphere Discussion, doi: 10.5194/tc-2017-87, 2017.

**General comments**

This paper reports a spatially integrated modelling and analysis framework with strong satellite data support, including surface temperature, snow coverage, depth, and density, and soil moisture. The authors also used airborne radar data for their analysis. This approach represents the new development and future directions of satellite based permafrost mapping and monitoring. With more satellite and airborne data available in the coming years with the support of NASA's ABoVE program, such an approach will greatly improve our capacity to understand and monitor the northern ecosystems and permafrost. The data description, sensitivity tests, and results analysis are clear and detailed, and the sources of uncertainties and limitations have been indicated in the discussion section. Such a new development and the preliminary results are worthy to be published.

**Major comments**

Actually, I do not have major concerns about this paper since it represents a new approach development with great potentials, rather than the final results and conclusions. With more and more data available from the ABoVE, such an approach (this framework or similar ones) will be improved gradually. With this, I suggest that the tone of the paper and language may put more attention on this forward looking and future development perspective. Before describing the model and input data in section 2.1, you may add a paragraph about the ideas and design of the overall framework.

Page 3, Line 24: "The soil process model was run at 1-km resolution and 8-day time step". The original model is run using a one-day time step implicitly. A time step of 8-day is too long. Clarify whether that is the case or you interpolated the 8-day land surface temperature data.

Page 6-7: SOC distribution in soil profile. Most northern lands have a pure organic layer (including lichen, mosses, peat or organic materials from leaf and roots) above the mixture of

mineral and organic matter, and this pure organic layer is very important for active layer thickness. Clarify whether you treatment of SOC distribution with depth reflected this phenomenon. The fraction of soil organic matter (SOM) content in a layer depends on the bulk density of the soil layer and the amount of SOM in this layer. So it would be useful to explain how do you estimate the bulk density of the soil layers and the general vertical distribution patterns of the fraction of SOM.

Page 14, Line 4, "after introducing a statistical distribution of the regional SOC spatial pattern". Put more explanation in the method section about this statistical distribution.

Figure 3. The model over estimated ALT at most CALM sites when permafrost probability is lower. It would be interesting if you can put some explanation about that systematic bias (due to model, input data, or spatial resolutions etc.?).

Figure 5 is not very clear, probably is not necessary.

Figures S2b and S2c: for easier comparison, is it better to use percentage of pixels in a latitude zone rather than number of pixels in the zone?

**Minor comments**

At some places, you used the phrases "soil active layer", "permafrost active layer". You may just say "active layer".

At many place, you used the phase "model ALT", it may be replaced by "modeled ALT". You also used "model ALT simulation" at many place. You may just use "modeled ALT".

Page 5, line 11: delete "first".

Page 6, line 5: "at October 2015", revised to "in October 2015".

---

## Referee Comment (RC2) · Anonymous Referee #2 · 2 Nov 2017

Title: Characterizing permafrost soil active layer dynamics and sensitivity to landscape spatial heterogeneity in Alaska Author(s): Yonghong Yi et al. MS No.: tc-2017-87

General Comments This is a well-written paper about the development of a model for Alaskan active layer thickness mapping based on satellite and ground data. The model was developed in response to a need to better understand spatial complexity (vertical and horizontal) of active layer thickness across varying biomes of Alaska. Complexities in the vertical distribution of SOC were most problematic/uncertain. Complexity and uncertainty remain, but improvements have been made in our understanding of how we might map active layer thickness across larger regions of the Arctic. I have no

major concerns about this paper, but would suggest addressing some of the comments below.

Specific Comments 1. P. 2 In the paragraph beginning with line 19 you state that information is lacking about the subsurface spatial variability of SOC, yet don't acknowledge others that have attempted to tackle aspects of this (although not Alaska). Suggest looking at Burnham and Sletten 2010 (doi:10.1029/2009GB003660) and similar refs within.

2. P. 2. Could you add a sentence between lines 27 and 29 stating the goal of the project?

3. I don't quite understand lines p.5 28-29 "The OLT was used to define the depth of the model soil layers with 100% SOC fraction." Does this mean you are assuming that 100% of the carbon extends only to the base of the OLT? Is this because you know you have Histels at these three tower sites? Was this for simplification of the parameterization process?

4. P. 7 line 2. While it helps simplify the analysis to assume an exponential vertical decay of SOC, this is not always the case for highly cryoturbated soils. In many cases there is more SOC at depth than at the surface (especially in patterned ground covered surfaces). I don't suggest changing this parameter, but do acknowledge that it can be highly variable for your higher latitude regions.

5. P. 8 paragraph beginning with line 18. Has bulk density been taken into account somewhere here when estimating SOC?

6. P. 14. It's unclear that getting a better estimate of subsurface SOC variability is ever really possible, but you could look to other arctic SOC studies showing that the most underestimated SOC stocks in the active layer are those found in the least vegetated arctic biomes.

7. Figure 8. A bit of clarification could be used in the caption. I'm unclear how uncertainty in 8a is represented by centimeters.

8. This is not absolutely necessary, but it would be good to add a note in the conclusion about the possibility (or not) of expanding this model to broader areas of the Arctic.

Technical Corrections 1. Very surprised to see that Arctic is misspelled in the first line of the abstract 2. P. 6 line 16, I don't believe that "in prep" submissions are considered acceptable references. 3. P. 7 line 17 I don't believe that "in review" submissions are acceptable either. 4. P. 13 line 13. Indicate instead of indicated 5. Minor point. Scale bar in Figure 3b appears taller from 70-100% than the rest of the bar. 6. Figure 5. I recommend not using blue and purple colors as they are not distinct enough from one another.

---

## Author Comment (AC1) · 7 Nov 2017

**Response to referee's comments on "Characterizing permafrost soil active layer dynamics and sensitivity to landscape spatial heterogeneity in Alaska"**

**Authors:** Y. Yi, J. S. Kimball, R. Chen, M. Moghaddam, R. H. Reichle, U. Mishra, D. Zona, W. C. Oechel

*Dear Editor,*
*We appreciate the helpful comments from the two reviewers. Our responses to the comments are provided in the following text, and the revised manuscript is enclosed as a supplement with changes highlighted. Thank you for considering our manuscript.*

**Review 1#:**

*1) I do not have major concerns about this paper since it represents a new approach development with great potentials, rather than the final results and conclusions. With more and more data available from the ABoVE, such an approach (this framework or similar ones) will be improved gradually. With this, I suggest that the tone of the paper and language may put more attention on this forward looking and future development perspective. Before describing the model and input data in section 2.1, you may add a paragraph about the ideas and design of the overall framework.*

**Response:**

Thank you for this suggestion. We added the following sentences in both the Introduction and conclusion sections to address our objectives and future development perspective:

Page 2, Line 29-34: "The on-going NASA Arctic-Boreal Vulnerability Experiment (ABoVE) field campaign is collecting a wide range of datasets intended to support regional integration and synthesis of geospatial information and associated data products generated from airborne and spaceborne remote sensing, and detailed ground observations. A major goal of the ABoVE is to develop a modelling framework to improve representation of key processes in the Arctic and boreal landscape, and study potential climate feedbacks via scaling of local processes to broader spatial extents using multi-scale and multi-sensor remote sensing (Goetz et al., 2011)."

Page 16, Line 12-15: "Additional AirMOSS/UAVSAR radar measurements will become available from the ABoVE airborne campaign in Alaska and western Canada, representing more extensive climate, terrain and vegetation conditions, and allowing for further testing and refinement of the modeling framework."

*2) Page 3, Line 24: "The soil process model was run at 1-km resolution and 8-day time step". The original model is run using a one-day time step implicitly. A time step of 8-day is too long. Clarify whether that is the case or you interpolated the 8-day land surface temperature data.*

**Response:**

The time step of the model was mostly constrained by the input datasets used in this study. The MODIS land surface temperature (LST) data was largely affected by clouds in the study area, and therefore, 8-day temporal composite LST was used as the model inputs. So is the MODIS snow cover (SCE) product. Our test run indicated relatively small differences between model simulated soil temperatures at 8-day and daily time steps as indicated below (e.g. Fig. R1). Here, the daily model simulations were derived using daily LST inputs interpolated from the coarser 8-day MODIS data, with daily snow depth and density inputs from MERRA reanalysis.

(a)

[Figure]

(b)

Fig. R1 Comparison of model test runs for pixel 149.3°W, 68.6°N at 8-day and daily time steps: (a) the model simulated annual time series of soil temperature at 8cm and 105cm; (b) the model simulated annual mean soil temperatures at 105cm during the 50-year spin-up process.

*3) Page 6-7: SOC distribution in soil profile. Most northern lands have a pure organic layer (including lichen, mosses, peat or organic materials from leaf and roots) above the mixture of mineral and organic matter, and this pure organic layer is very important for active layer thickness. Clarify whether you treatment of SOC distribution with depth reflected this phenomenon. The fraction of soil organic matter (SOM) content in a layer depends on the bulk density of the soil layer and the amount of SOM in this layer. So it would be useful to explain how do you estimate the bulk density of the soil layers and the general vertical distribution patterns of the fraction of SOM.*

**Response:**

We agree that surface organic soils are very important for soil temperature and ALT simulations in the boreal and Arctic region, which is also highlighted in our conclusions. We added the

following sentences to clarify how we calculated the soil organic fraction for each soil layer up to ~1m below surface based on the SOC inventory data:

Page 5, Line 23-27: "The SOC data was first distributed through the top 10 model layers ($\leq$1.05 m depth) following an exponentially decreasing curve (Jobbagy and Jackson, 2000; Hossain et al, 2015) to calculate the soil carbon fraction for each soil layer. Different decreasing rates were assigned for the boreal forest and other biome types as described in Section 2.3.1. The soil physical properties for each soil layer were assumed to be a weighted combination of values of mineral soils and pure organic soils based on the estimated soil carbon fraction following Yi et al. (2015)."

Page 7, Line 22-27: "The soil carbon or organic fraction for each soil layer was estimated as:

$$f_{sc,i} = SOCC(z_i)/SOCC_{max} \tag{2}$$

Where $SOCC(z_i)$ is the estimated soil carbon density at the centre depth ($z_i$) of soil layer i and $SOCC_{max} = 130$ kg m$^{-3}$ is the maximum soil carbon density of peat soils (Farouki, 1981). Mineral soils may also contain a high soil carbon density but low soil organic fraction due to much higher bulk density. Therefore, the soil carbon fraction was adjusted based on an empirical relationship between soil carbon concentration and bulk density (Hossain et al., 2015) when the SOCC is below 40 kg m$^{-3}$."

*4) Page 14, Line 4, "after introducing a statistical distribution of the regional SOC spatial pattern". Put more explanation in the method section about this statistical distribution.*

**Response:**

In the methods (Section 2.3.2), we indicated the statistical distribution of regional SOC was following the statistical distribution of radar retrieved soil moisture content in the zone between 69.5 °N and 70 °N due to a generally close relationship between SOC content and local soil wetness variability (Mishra and Riley, 2015). This statistical distribution was also similar to the field data as described in Page 9, Line 17-20 of the revised paper:

"Specifically, the Run3 scenario assumes the regional SOC distribution follows the statistical distribution of radar retrieved soil moisture across the DH sub-region from 69.5-70°N (Figure S2 c), resulting in an estimated SOC range from 21 to 69 kgC m$^{-2}$, and a mean value of 45 kgC m$^{-2}$. This statistical distribution was similar to the OLT distribution observed from field sampling data in northern Canada (Zhang et al., 2014)."

*5) Figure 3. The model over estimated ALT at most CALM sites when permafrost probability is lower. It would be interesting if you can put some explanation about that systematic bias (due to model, input data, or spatial resolutions etc.?).*

**Response:**

The larger apparent discrepancy between model estimated ALT at 1-km grid scale and the point-scale in-situ measurements for CALM sites with lower permafrost probability larger spatial heterogeneity in permafrost and active layer conditions in these areas, which may not be adequately represented by the coarse model resolution and associated drivers. For example, the permafrost probability map based on soil inventory data (Fig. 2b) showed lower probability of near-surface (<1m) permafrost occurrence in those areas while the in-situ data indicated ALT generally<1m. An alternative and potentially more effective model approach would be to provide model estimates of near-surface permafrost probability and the frequency distribution of ALT, rather than deriving a single ALT value for each 1-km$^2$ grid cell in those areas. The following explanation was added in the Section 3.1.1 (Page 10, Line 19-24) of the revised manuscript:

"Larger differences between model simulations and in-situ ALT measurements in areas with lower permafrost probability is not unexpected due to strong surface heterogeneity in permafrost conditions, leading to larger discrepancy between model simulations representing a single ALT value for each 1-km2 grid cell and the point-scale measurements. For example, the in situ CALM site measurements indicated ALT generally shallower than 1m (Fig. 3d), while the satellite and soil inventory based PP map showed a lower probability of permafrost occurrence within 1-m surface soils in those areas."

*6) Figure 5 is not very clear, probably is not necessary.*

**Response:**

We revised the figure to make the colors more distinct, and moved it to the Supplement section (Fig. S8).

[Figure]

**Figure S8: Comparisons of model simulated (dashed lines) and in-situ observed (solid lines) ALT time series at the three in-situ CALM sites located within the DH sub-region (Figure 4). Both the model simulations and in-situ observations at**

**the Franklin Bluff site show similar inter-annual variability as the Deadhorse site, and are thus not shown. The R values are the correlations between the in-situ observed and model ALT estimates from 2001 to 2015.**

*7) Figures S2b and S2c: for easier comparison, is it better to use percentage of pixels in a latitude zone rather than number of pixels in the zone?*

**Response:**

We revised Fig. S2 (b) and (c) and used percentage of pixels for each latitude zone in the y axis.

[Figure]

**Figure S2: Regional ALT and soil moisture (SM) spatial distributions derived from AirMOSS P-band and UAVSAR L-band radar backscatter retrievals within the Alaska Dalton Highway (DH) sub-region acquired in October 2015: (a) the radar retrieved volumetric SM (m³ m⁻³) of the soil active layer; the locations of in-situ CALM site are denoted by black stars. (b-c): the frequency distribution of the local scale (50-m resolution) ariborne radar based ALT (c) and active layer SM (c) estimates for different latitudinal bands within the DH sub-region.**

*8) At some places, you used the phrases "soil active layer", "permafrost active layer". You may just say "active layer". At many place, you used the phase "model ALT", it may be replaced by "modeled ALT". You also used "model ALT simulation" at many place. You may just use "modeled ALT".*

**Response:**

We now use "active layer" and "modeled ALT" throughout the paper following reviewer recommendations.

*9) Page 5, line 11: delete "first". Page 6, line 5: "at October 2015", revised to "in October 2015".*

**Response:**

They were corrected.

**Review 2# (Remarks to author):**

*1) P. 2 In the paragraph beginning with line 19 you state that information is lacking about the subsurface spatial variability of SOC, yet don't acknowledge others that have attempted to tackle aspects of this (although not Alaska). Suggest looking at Burnham and Sletten 2010 (doi:10.1029/2009GB003660) and similar refs within.*

**Response**:

In the introduction, we mostly focused on the lack of information on spatial variability of active layer conditions, rather than SOC variability (though this is a key variable affecting ALT). But we recognize that there have been a number of studies investigating the spatial variability of subsurface SOC in the Arctic and boreal region, notably Ping et al. (2008) and Burnham and Sletten (2010). We added the following sentences to address this:

Page 2, Line 18-20: "Particularly, large uncertainties remain in characterizing regional variability of subsurface soil organic carbon (SOC) due to limited ground observations of this parameter in the Arctic region (Ping et al., 2008; Burnham and Sletten, 2010) and its effects on ground temperature evolution."

*2) P. 2. Could you add a sentence between lines 27 and 29 stating the goal of the project?*

**Response**:

We added the following sentences in the Introduction to address the goal of this project:

Page 2, Line 29-34: "The on-going NASA Arctic-Boreal Vulnerability Experiment (ABoVE) field campaign is collecting a wide range of datasets intended to support regional integration and synthesis of geospatial information and associated data products generated from airborne and spaceborne remote sensing, and detailed ground observations. A major goal of the ABoVE is to develop a modelling framework to improve representation of key processes in the Arctic and boreal landscape, and study potential climate feedbacks via scaling of local processes to broader spatial extents using multi-scale and multi-sensor remote sensing (Goetz et al., 2011).

Consistent with the ABoVE modelling goal..."

*3) I don't quite understand lines p.5 28-29 "The OLT was used to define the depth of the model soil layers with 100% SOC fraction." Does this mean you are assuming that 100% of the carbon extends only to the base of the OLT? Is this because you know you have Histels at these three tower sites? Was this for simplification of the parameterization process?*

**Response**:

We made this assumption due to a lack of information on the in-situ soil characteristics and the need for a simplified model parameterization scheme. Moreover, even though there may be a certain amount of soil organic carbon in the mineral soil below the surface organic soil, its soil carbon concentration is generally much lower than the organic soils due to much higher bulk density of mineral soils (Hossain et al., 2015). Therefore, the "surface" organic soils were deemed to be more important in controlling soil thermal conditions within the active layer. We clarified this in the text:

Page 6, Line 5-7: "The OLT observations were used to define the depth of the model soil layers with 100% SOC fraction; this simplifying assumption was made in the absence of more detailed SOC profile measurements and to facilitate the model parameterization process."

*4) P. 7 line 2. While it helps simplify the analysis to assume an exponential vertical decay of SOC, this is not always the case for highly cryoturbated soils. In many cases there is more SOC at depth than at the surface (especially in patterned ground covered surfaces). I don't suggest changing this parameter, but do acknowledge that it can be highly variable for your higher latitude regions.*

**Response**:

We agree with the reviewer that a large amount of SOC can be transferred from surface to depth due to cryoturbation in the permafrost region. This will add additional uncertainty to the SOC profile generated in this study. However, the SOC concentration or fraction generally decreased much more rapidly than the total SOC content due to the increase in soil density with depth. We addressed this uncertainty in the discussion section of the revised paper:

Page 14, Line 28-32: "The SOC content was assumed to decrease exponentially with increasing depth from the surface (Eq. 1), which may significantly underestimate the SOC of deep soils in areas strongly affected by cryoturbation (Ping et al., 2008; Burnham and Sletton, 2010). However, this process may have a relatively limited effect on the estimated soil carbon fraction due to general increases in soil bulk density and thus lower soil carbon concentration with depth (Hossain et al., 2015)."

*5) P. 8 paragraph beginning with line 18. Has bulk density been taken into account somewhere here when estimating SOC?*

**Response**:

The soil carbon fraction of each soil layer up to 1m below surface was estimated based on the SOC content and maximum SOC density (130 kg m$^{-3}$) assumed for peat soils. The soil carbon fraction was then adjusted using an empirical relationship between SOC concentration and bulk density derived from field data in northern Canada when the SOC content is below 40 kg m$^{-3}$. We clarified this in the methods section of the revised paper:

Page 7, Line 22-27: "The soil carbon or organic fraction for each soil layer was estimated as:

$$f_{sc,i} = SOCC(z_i)/SOCC_{max} \qquad (2)$$

Where $SOCC(z_i)$ is the estimated soil carbon density at the centre depth ($z_i$) of soil layer i and $SOCC_{max} = 130$ kg m$^{-3}$ is the maximum soil carbon density of peat soils (Farouki, 1981). Mineral soils may also contain a high soil carbon density but low soil organic fraction due to much higher bulk density. Therefore, the soil carbon fraction was adjusted based on an empirical relationship between soil carbon concentration and bulk density (Hossain et al., 2015) when the SOCC is below 40 kg m$^{-3}$."

*6) P. 14. It's unclear that getting a better estimate of subsurface SOC variability is ever really possible, but you could look to other arctic SOC studies showing that the most underestimated SOC stocks in the active layer are those found in the least vegetated arctic biomes.*

**Response**:

We agree that it is very challenging to accurately estimate the subsurface SOC variability due to very sparse data and strong landscape heterogeneity in the Arctic. Previous studies may also significantly underestimate the SOC in sparse vegetated arctic biomes. For example, Burnham and Sletten (2010) pointed out that previous studies underestimated the SOC in polar desert and semidesert regions much more than more densely vegetated regions in the High Arctic. However, the relatively large bias was due to a very low total SOC content in those areas reported from previous studies (e.g. 0.046 kg/m$^2$ and 2.19 kg/m$^2$ for polar semidesert and desert region in Bliss and Matveyeva 1992). The SOC content for those biomes is generally low and will have relatively small impact on the model estimated soil thermal conditions.

*7) Figure 8. A bit of clarification could be used in the caption. I'm unclear how uncertainty in 8a is represented by centimeters.*

**Response**:

We added the following sentences in the caption of Figure 7 (replacing the original Figure 8) to clarify this:

"The ALT uncertainties were calculated as the standard deviation between the model baseline simulations and the sensitivity runs driven by adding uncertainties in the regional SOC map, soil moisture and snow density data (Figure 1)."

The uncertainty in modelled ALT generally increases with the mean ALT estimated by the model baseline simulations.

*8) This is not absolutely necessary, but it would be good to add a note in the conclusion about the possibility (or not) of expanding this model to broader areas of the Arctic.*

**Response**:

We do plan to further refine our modeling framework and apply to the NASA ABoVE domain with potential extension to the pan-Arctic region (albeit with coarser resolution):

Page 16, Line 12-15: "Additional AirMOSS/UAVSAR radar measurements will become available from the ABoVE airborne campaign in Alaska and western Canada, representing more extensive climate, terrain and vegetation conditions, and allowing for further testing and refinement of the modeling framework."

*9) Very surprised to see that Arctic is misspelled in the first line of the abstract*

**Response**:

It was corrected.

*10) P. 6 line 16, I don't believe that "in prep" submissions are considered acceptable references. 3. P. 7 line 17 I don't believe that "in review" submissions are acceptable either.*

**Response**:

We removed the submission "in prep" and replaced the "in review" reference with a published citation (Reichle et al., 2017) in the revised paper.

*11) P. 13 line 13. Indicate instead of indicated. Scale bar in Figure 3b appears taller from 70-100% than the rest of the bar.*

**Response**:

They were corrected.

*12) Figure 5. I recommend not using blue and purple colors as they are not distinct enough from one another.*

**Response**:

We revised the figure to make the colors more distinct, and also moved the figure to the Supplement (Fig. S8).

**References:**

1. Bliss, L. C., and N. V. Matveyeva (1992), Circumpolar Arctic vegetation, in Arctic Ecosystems in a Changing Climate, edited by F. S. Chapin III et al., pp. 59–89, Academic, San Diego, Calif.

2. Burnham, J. H. and Sletten, R. S.: Spatial distribution of soil organic carbon in northwest Greenland and underestimates of high Arctic carbon stores, Global Biogeochemical Cycles, 24, 2010.

3. Ping, C. L., Michaelson, G. J., Jorgenson, M. T., Kimble, J. M., Epstein, H., Romanovsky, V. E., and Walker, D. A.: High stocks of soil organic carbon in the North American Arctic region, Nature Geoscience, 1, 615-619, 2008.

4. Reichle, R. et al.: Assessment of the SMAP Level-4 Surface and Root-Zone Soil Moisture Product Using In Situ Measurements, Journal of Hydrometeorology, 18, 2621-2645, 2017.

---

## Editor Decision (ED1)

Dear Dr. Yi,

Thank you for your revised manuscript "Characterizing permafrost soil active layer dynamics and sensitivity to landscape spatial heterogeneity in Alaska" (tc-2017-87 version 3). You have addressed the reviewer's comments and with minor revisions the manuscript will be ready for publication.

P1 L 15: "~50 m" becomes "~ 50 m resolution"

P1 L26: "permafrost active layer conditions." AL is by definition above Pf. Also, it would be good to add some text about the framework and future development perspective. Sentence becomes something like "active layer conditions and refinement of the modelling framework across a larger domain."

P3 L31: After the first sentence in this paragraph, please include a line about the small differences you fine between the 8-day and daily time steps. Add Figure R1 to your supplementary materials, and refer to it. The caption for the figure will include a component of your response to Reviewer 1.

P6 L14: "backscatter measurements acquired" becomes "backscatter measurements (~ 50 m resolution) acquired"

P7 L2: "SM and snow density" becomes "SM, and snow density"

P7 L9: high, baseline and low SOC" becomes "high, baseline, and low SOC"

P8 L30: "~1km" becomes "~ 1 km"

P9 L23 and throughout: Search and replace "in-situ" with "in situ"

P11 L31: "accumulated degree" becomes "accumulated thawing degree"

P11 L3: "MODIS LST degree" becomes "MODIS LST thawing degree"

P12 L8: "MODIS LST degree" becomes "MODIS LST thawing degree"

P13 L18: "indicated" becomes "indicate"

P14 L11: Please cite Zhang, T.: Influence of the seasonal snow cover on the ground thermal regime: An overview, Reviews of Geophysics, 43, RG4002, 2005.

P14 L18: "(50-m)" becomes "(~ 50 m)"

P15 L15: "(Due et al. 2015; Bartsch et al. 2016)" becomes "(Due et al., 2015; Bartsch et al., 2016)"

P15 L21: "(Liston and Sturm 1998, Gisnas et al 2016)" becomes "(Liston and Sturm, 1998; Gisnas et al., 2016)"

P15 L27: "(Jorgenson et al 2006, Osterkamp et al 2009, Grosse et al 2011)" becomes "(Jorgenson et al., 2006; Osterkamp et al., 2009; Grosse et al., 2011)"

P15 L32: "(~50m resolution)" becomes "(~ 50 m resolution)"

References: Carefully review and write out all journal titles that have been abbreviated.

Supplementary materials: Please add in references cited beneath Table S1 and in the Caption for Figure S3.

Throughout the text, tables, figures and captions, both in the main document and the supplementary material: Carefully ensure that units are reported in a similar manner. In many cases units are reported as cm, but in others units are reported correctly in m (Page 13 Lines 18 to 24 are a good example of this). Some figures show units in m, but most report units in cm. I suggest changing cm to m and centimetre to metre throughout, adjusting the reported numerical valuse accordingly. I recognize that this is tedious, but will make for a better read.

Throughout the text, tables, figures and captions, both the main document and the supplementary material: Thin spaces should be used before and after the following mathematical symbols: $\pm$, $=$, $<$, $>$, $\leq$, $\geq$, $+$, $-$, $\div$, $\times$, $\cdot$, $\approx$, $\sim$, $\cap$, $\int$, $\prod$, $\sum$, and $|$.

Throughout the text, tables, and figures, both the main document and the supplementary material: Thin spaces should also the unit symbol from the numerical value, and this applies to the degree Celsius (e.g., "10 °C"). The exception is for degree, minute, and second for plane angle (e.g., "coarse resolution ($\sim 5°$)"). Please carefully review the text, figures, captions, and tables in the main document and the supplementary material and correct as necessary.

Throughout the text, tables, and figures, both the main document and the supplementary material: For consistency, please do not use mixed units. E.g., change 10 m/s to 10 m s$^{-1}$. This notation change is most often required in figures for such things as axis titles.

---

## Author Response (AR2)

**Response to editor's comments on "Characterizing permafrost soil active layer dynamics and sensitivity to landscape spatial heterogeneity in Alaska"**

**Authors:** Y. Yi, J. S. Kimball, R. Chen, M. Moghaddam, R. H. Reichle, U. Mishra, D. Zona, W. C. Oechel

*Dear Editor,*

*Thank you for your careful reading and consideration of our manuscript. We have addressed your comments in the manuscript with changes are highlighted.*

*1) P1 L 15: "~50 m" becomes "~ 50 m resolution"*

**Response:** this was corrected.

*2) P1 L26: "permafrost active layer conditions." AL is by definition above Pf. Also, it would be good to add some text about the framework and future development perspective. Sentence becomes something like "active layer conditions and refinement of the modelling framework across a larger domain."*

**Response:** We revised the sentence based on your suggestions:

Page 1, Line 26-27: "... will enable more accurate predictions of active layer conditions and refinement of the modeling framework across a larger domain."

*3) P3 L31: After the first sentence in this paragraph, please include a line about the small differences you fine between the 8-day and daily time steps. Add Figure R1 to your supplementary materials, and refer to it. The caption for the figure will include a component of your response to Reviewer 1.*

**Response:** We added a sentence explaining why we chose the 8-day time step for the modeling:

Page 4, Line 3-5: "The MODIS LST and SCE data were largely affected by clouds in the study area, and therefore 8-day temporal composite data were used as the model inputs. Our test runs indicated relatively small differences between model simulated soil temperatures at 8-day and daily time steps (Figure S1)."

*4) P6 L14: "backscatter measurements acquired" becomes "backscatter measurements (~ 50 m resolution) acquired"*
*P7 L2: "SM and snow density" becomes "SM, and snow density"*
*P7 L9: high, baseline and low SOC" becomes "high, baseline, and low SOC"*
*P8 L30: "~1km" becomes "~ 1 km"*
*P9 L23 and throughout: Search and replace "in-situ" with "in situ"*
*P11 L31: "accumulated degree" becomes "accumulated thawing degree"*
*P11 L3: "MODIS LST degree" becomes "MODIS LST thawing degree"*
*P12 L8: "MODIS LST degree" becomes "MODIS LST thawing degree"*
*P13 L18: "indicated" becomes "indicate"*

**Response:** all these were corrected.

*5) P14 L11: Please cite Zhang, T.: Influence of the seasonal snow cover on the ground thermal regime: An overview, Reviews of Geophysics, 43, RG4002, 2005.*

**Response:** this was cited.

*6) P14 L18: "(50-m)" becomes "(~ 50 m)"*

**Response:** this was corrected.

*7) P15 L15: "(Due et al. 2015; Bartsch et al. 2016)" becomes "(Due et al., 2015; Bartsch et al., 2016)"*
  *P15 L21: "(Liston and Sturm 1998, Gisnas et al 2016)" becomes "(Liston and Sturm, 1998; Gisnas et al., 2016)"*
  *P15 L27: "(Jorgenson et al 2006, Osterkamp et al 2009, Grosse et al 2011)" becomes "(Jorgenson et al., 2006; Osterkamp et al., 2009; Grosse et al., 2011)"*

**Response:** the citations were corrected.

*8) P15 L32: "(~50m resolution)" becomes "(~ 50 m resolution)"*

**Response:** this was corrected.

*9) References: Carefully review and write out all journal titles that have been abbreviated.*

**Response:** the citations were updated with journal titles spelled out.

*10) Supplementary materials: Please add in references cited beneath Table S1 and in the Caption for Figure S3.*

**Response:** the references were added.

*11) Throughout the text, tables, figures and captions, both in the main document and the supplementary material: Carefully ensure that units are reported in a similar manner. In many cases units are reported as cm, but in others units are reported correctly in m (Page 13 Lines 18 to 24 are a good example of this). Some figures show units in m, but most report units in cm. I suggest changing cm to m and centimetre to metre throughout, adjusting the reported numerical values accordingly. I recognize that this is tedious, but will make for a better read.*

**Response:** Since many of the units for ALT simulations were reported in cm with 2 decimal places, we changed the units from "m" to "cm" in order not to lose the accuracy. We now report the ALT results in cm or cm $yr^{-1}$ (trends) throughout the manuscript, figures, tables and supplement material.

*12) Throughout the text, tables, figures and captions, both the main document and the supplementary material: Thin spaces should be used before and after the following mathematical symbols.*

*Throughout the text, tables, and figures, both the main document and the supplementary material: Thin spaces should also the unit symbol from the numerical value, and this applies to the degree Celsius (e.g., "10 °C"). The exception is for degree, minute, and second for plane angle (e.g., "coarse resolution (~ 5°)"). Please carefully review the text, figures, captions, and tables in the main document and the supplementary material and correct as necessary.*

*Throughout the text, tables, and figures, both the main document and the supplementary material: For consistency, please do not use mixed units. E.g., change 10 m/s to 10 m s$^{-1}$. This notation change is most often required in figures for such things as axis titles.*

**Response:** All these have been corrected throughout the manuscript, figures, tables and also supplementary material.